# Data-Efficient Operator Learning via Unsupervised Pretraining and In-Context Learning

**Wuyang Chen**\*
Simon Fraser University

**Jialin Song**\*
Simon Fraser University

**Pu Ren**
Lawrence Berkeley National Laboratory

**Shashank Subramanian**
Lawrence Berkeley National Laboratory

**Dmitriy Morozov**
Lawrence Berkeley National Laboratory

**Michael W. Mahoney**
International Computer Science Institute
Lawrence Berkeley National Laboratory
University of California, Berkeley

## Abstract

Recent years have witnessed the promise of coupling machine learning methods and physical domain-specific insights for solving scientific problems based on partial differential equations (PDEs). However, being data-intensive, these methods still require a large amount of PDE data. This reintroduces the need for expensive numerical PDE solutions, partially undermining the original goal of avoiding these expensive simulations. In this work, seeking data efficiency, we design unsupervised pretraining for PDE operator learning. To reduce the need for training data with heavy simulation costs, we mine unlabeled PDE data without simulated solutions, and we pretrain neural operators with physics-inspired reconstruction-based proxy tasks. To improve out-of-distribution performance, we further assist neural operators in flexibly leveraging a similarity-based method that learns in-context examples, without incurring extra training costs or designs. Extensive empirical evaluations on a diverse set of PDEs demonstrate that our method is highly data-efficient, more generalizable, and even outperforms conventional vision-pretrained models. We provide our code at `https://github.com/delta-lab-ai/data_efficient_nopt`.

## 1 Introduction

Recent advancements in machine learning methodology have shown promise in solving partial differential equations (PDEs) [24, 67, 38, 45, 44, 50, 39, 25]. A significant development in this area is the concept of operator learning for PDEs. This approach differs from traditional neural network methods, which are restricted to fixed-dimension input and output, since neural operators focus on learning mappings between function spaces [38, 45, 44]. Like other neural network methods, neural operators are recognized to be universal approximators for any continuous operator [46, 38], enabling them to approximate any physical operator, including solution operators for various parametric PDE families. A solution operator is defined as a function that maps physical inputs to output solutions. Previous work has shown that in simple settings, neural operators can effectively capture complex, multi-scale dynamic processes [45, 46, 74, 73].

However, neural operators tend to suffer from a problem common to other deep networks, namely the *need for enormous quantities of data*. Limited availability of data is common in science and

---

\*Equal contribution.

engineering. High-fidelity numerical simulations are computationally costly or even infeasible for many applications [71]. For example, an extreme-scale simulation of magnitude 7.0 earthquake at frequencies up to 10 Hz in San Francisco requires 3600 Summit GPU nodes and 42.7 hour [53].

Motivated by this data-efficiency challenge, recent works, particularly in natural language processing (NLP) and computer vision (CV), have focused on unsupervised (or self-supervised) pretraining[2] to reduce the cost of collecting or generating labeled data. Such pretrained models have been shown to be highly data-efficient in downstream fine-tuning [7, 29, 8], and they can even become few-shot learners without any downstream data [3]. In CV, researchers collect large amounts of natural images without any manual labels, and then pretrain visual encoders with proxy tasks, such as Noise-Contrastive Estimation (NCE) [59], masked reconstruction [28], rotation and jigsaw prediction [58, 20]. In NLP, people typically pretrain models via next-word prediction or masked tokens [3, 12].

However, unsupervised pretraining is still largely underexplored in Scientific Machine Learning (SciML). Therefore, our core question is: *How can we design unsupervised pretraining for operator learning to reduce the data simulation costs?*

In this work, we resort to unsupervised pretraining for neural operator learning to achieve data efficiency in SciML. We overview our framework in Figure 1. First, we define unlabeled data for PDEs, which avoids heavy computation costs for simulating PDE solutions. We propose two physics-inspired reconstruction-based proxy tasks, and we pretrain neural operators on unlabeled PDE data. We demonstrate that with unsupervised pretraining, our neural operators not only improve counterparts trained with more simulated data, but also they outperform off-the-shelf pretrained checkpoints from other popular domains (such as CV) that are ready for fine-tuning. Then, to further improve the data efficiency during out-of-distribution (OOD) inference, we design a similarity-based method that learns in-context examples [3, 49, 80, 81, 47]. This approach introduces zero overhead during training: one just maintains the standard training pipeline, and it can be seamlessly plugged in for OOD inference, without further fine-tuning. In more detail, we summarize our main contributions:

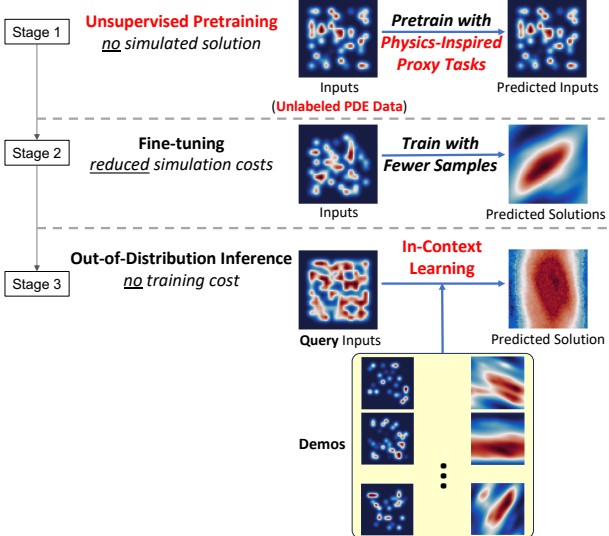

Figure 1: Overview of our framework for data-efficient neural operator learning (with our contributions highlighted in red). Stage 1: Unsupervised pretraining only on unlabeled PDE data. Stage 2: Fine-tuning with reduced simulation costs of PDE data. Stage 3: Test-time in-context examples can improve the neural operator's out-of-distribution performance, without additional training costs.

1. We introduce unlabeled PDE data and unsupervised pretraining for data-efficient neural operator learning. We show that our method can achieve better performance than models trained with more simulated PDE solutions, or fine-tuned from public checkpoints pretrained on other benchmarks, demonstrating the importance of unsupervised pretraining on domain-specific PDE data.

2. We propose a similarity-based method to improve the OOD generalization of neural operators, which is flexible and can scale up to a large number of unseen in-context examples ("demos").

3. We provide detailed empirical evaluations on both diverse PDE benchmarks and also several real-world scenarios, demonstrating that we can achieve both strong forward modeling performance and significant savings in PDE simulations.

---

[2]In this paper, we use the terms "unsupervised pretraining" and "self-supervised pretraining" interchangeably.

## 2 Related Works

### 2.1 Machine Learning for Scientific Modeling

There has been a long history of using learning-based methods to model physical scientific phenomena [41, 42, 6, 5]. A representative line of work is so-called physics-informed neural networks (PINNs) [67, 87, 19, 18, 69], which try to incorporate physics in neural networks by including the differential form of the PDE as an additional physics loss regularization term. However, this paradigm is confined to specific PDE scenarios (e.g., fixed PDE coefficients), instead of being more physics-agnostic. Moreover, recent work has highlighted several fundamental "issues" with PINN-based methods [40, 15]. On the other hand, operator learning methods, including Fourier Neural Operators [45, 44, 38] and Deep Operator Network [50], have achieved progress in approximating the solution operators of PDEs. Although these data-driven approaches show promise in learning PDE solutions, they (like many neural network-based methods) rely on vast quantities of high-fidelity labeled data. For operator learning, such data are usually computationally expensive to simulate [67, 2, 84]. More recently, people have tried to generate synthetic PDE solutions to train SciML models [27]. In contrast, our method works on unlabeled PDE data. This approach is distinct from the generation of synthetic PDE data, and it could be further combined as a semi-supervised learning strategy.

### 2.2 Unsupervised Pretraining and Foundation Models

Unsupervised (or self-supervised) pretraining is a key method in CV and NLP to achieve meaningful representations [7], data-efficient fine-tuning [31], and foundation models [1]. In CV, contrastive learning learns meaningful features by distinguishing between similar (positive) and different (negative) samples [59, 76, 7, 56, 29]. Masked Autoencoder (MAE) [28] uses a reconstructive approach where parts of the input are masked and the model learns to predict masked parts. In NLP, among the most prominent works are large language models (LLMs) such as GPT [3, 64, 65] and BERT [12, 66], which leverage token predictions for pretraining. Similar directions also show progress in SciML. For example, [2] and [54] propose to create augmented views in the solution space via Lie Symmetries; [73] study the scaling behavior of supervised pretraining and OOD generalization, charting directions for foundational models for SciML; [43] target learning astronomical foundation models with cross-modal contrastive learning; and [52] build large task-agnostic models with a broad understanding of common physical behavior to serve as foundation models for SciML.

### 2.3 In-Context Learning (ICL)

In-context learning (ICL) is a promising paradigm that helps deep networks generalize to unseen domains with a few in-context examples. Early works in CV seek to learn feature-level correspondence between the target and a few "shots," such that models can generalize to open-set unseen objects [17, 83]. In NLP, people find LLMs are naturally few-shot learners [3], and thus tuning or optimizing prompts becomes extremely important to improve the in-context learning performance of LLMs [85, 72]. More recently, within SciML, a different operator learning strategy, termed "in-context operator learning," has been proposed [80, 81, 47]. During both training and inference, the neural operator is asked to make predictions by explicitly leveraging a predefined number of so-called "demo" examples (pairs of physical parameters and simulated solutions). This approach provides a balance between model generalization and addressing data scarcity in the scenario of OOD testing.

## 3 Methods

In this section, we introduce our framework (outlined in Figure 1). We propose first to pretrain the model with unsupervised pretraining (Sec. 3.1), which will contribute to the data efficiency and reduced PDE simulation costs during standard training of neural operators. When we move to OOD scenarios during inference, we test our models with in-context examples (Sec. 3.2) to avoid further fine-tuning costs.

### 3.1 Unsupervised Pretraining

The core idea in unsupervised (or self-supervised) pretraining is to train a neural network with properly designed proxy tasks. These proxy tasks do not require labeled data, but they are designed

to be highly related to the supervised learning objectives of interest. While popular in CV and NLP, unsupervised pretraining on unlabeled data has never been explored in PDE operator learning in SciML. This is due to *two unresolved questions*: 1) What kinds of unlabeled data can we use to train neural operators? 2) How to design proxy tasks for PDE data? We address these questions below.

### 3.1.1 Unlabeled PDE Data

**How to Define Unlabeled PDE Data?**   In general, when a neural operator is trained on PDE datasets [45, 74, 73], it learns to map inputs (physical parameters, coordinates, forcing functions, initial conditions, etc.) to PDE solutions. Therefore, given a set of PDE data (collected via simulations or observations), its unlabeled version is defined as the one without PDE solutions. Our unlabeled PDE data is a broader concept of related inputs in modeling PDE systems. Let us consider the second-order linear differential equation as a general example. It is formulated as

$$\sum_{i,j=1}^{n} a_{ij}(x)u_{x_i x_j} + \sum_{i=1}^{n} b_i(x)u_{x_i} + c(x)u = f(x),$$

where $x \in \mathbb{R}^n$ represents physical space that varies in different systems (e.g. $n = 3$ for 2D time-dependent PDEs); the coefficients (or physical parameters) $a_{ij}, b_i, c$ are known from the physical process; $u$ is the target solution; and $f$ denotes an external forcing function [57]. We can consider two situations where solutions are unavailable:

- Time-independent equations: following [45, 73], our unlabeled PDE data include physical parameters $(a_{ij}, b_i, c)$, forcing functions ($f$), and coordinates (grids of the discrete physical space).
- Time-dependent equations (e.g., forecasting problems [74, 52]): without simulating the temporal dynamics, our unlabeled PDE data only include initial snapshot $u_0(x)$ that defines PDE systems. Note that collecting snapshots with temporal dynamics in large-scale scenes is more complex than capturing individual snapshots. For example, weather forecasting [30] and smoke dispersion [14] require continuous monitoring and multiple sensors, whereas single measurements are simpler and less resource-intensive. Long-term data collection often involves extensive networks and processing, unlike one-time measurements.

For concrete examples of different PDEs and unlabeled data that we will study, see Appendix A. There are two main reasons for pretraining only on unlabeled PDE data, as discussed below.

**Cheap Generation of Unlabeled PDE Data.**   One critical reason that leads to expensive computational costs when collecting PDE data is the time marching scheme [23, 22] in numerical simulation. However, only generating unlabeled PDE data and snapshots without temporal dynamics will be much cheaper than simulating solutions (Table 4), making our unsupervised pretraining highly feasible in practice. Our pretraining strategy will be very data-efficient, and can avoid the heavy computational cost of simulating complex time-dependent equations for massive high-fidelity labeled solutions.

**Benefits of Pretraining on Unlabeled PDE Data.**   Beyond the cheap generation, pretraining on unlabeled PDE data has the following benefits. First, *regularization against overfitting*. Unsupervised pretraining can strongly regularize the model towards better generalization. Second, *faster convergence*. Pretraining on unlabeled PDE data will provide neural operators with domain-adapted initializations and can accelerate training speed. Third, *meaningful representations*. Pretraining on unlabeled PDE data can help models extract useful representations for subsequent operator learning. We defer our results and experimental details in Figure 4 in Sec. 4.1.

### 3.1.2 Proxy Tasks

To illustrate our general approach of constructing proxy tasks, we choose two variants of reconstruction as our core proxy tasks. In particular, we will input unlabeled PDE data to our neural operators, and after a decoder network, we will force the output to be close to the input. We consider two perturbation variants (or augmented views). They are inspired by real-world settings when people collect scientific data, and they are important invariances we need to introduce to SciML models.

**Masked Autoencoder.**   Masked autoencoders (MAEs) have been shown to be scalable self-supervised learners [28]. The method is conceptually simple: remove a portion of the input data, and

learn to predict the removed content. These methods enable training in NLP and CV of generalizable models containing over one hundred billion parameters [12, 3]. Here, we investigate the potential of MAE for scientific modeling.

Motivation: PDE dynamics are invariant to sparse sensing of the full field scientific data. It is very common [4, 16, 32] that scientific data need to be collected from sparse sensors, and that people need to reconstruct or generate data for domains without sensors. We enforce our model to learn sensor invariance via random masking, and we extract the invariant features over distorted views of the same unlabeled PDE data. The invariance to sparse sensing of scientific data will facilitate the robustness of the representations from MAE.

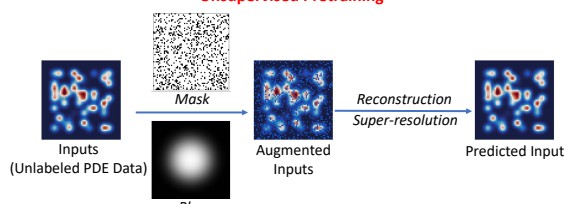

Figure 2: Overview: unsupervised pretraining via MAE and super-resolution. During pre-training, in the input unlabeled PDE data, a random subset (e.g., 70%) of spatial locations are masked, followed by a Gaussian blur. After the encoder and decoder, the full set of input is required to be reconstructed.

Therefore, we consider MAE as a proxy task. Specifically, our MAE is a straightforward autoencoding approach that reconstructs the original signal, given its partial observation. Like all autoencoders, our approach has an encoder that maps the observed signal to a latent representation, and a decoder that reconstructs the original signal from the latent representation. We randomly sample masks according to a certain masking ratio (i.e., the ratio of removed areas), and the values of masked areas of unlabeled PDE data are set to zero. Our loss function computes the mean squared error (MSE) between the reconstructed and original input. We compute the loss only on masked areas; if no mask is applied (i.e., a vanilla autoencoder), the MSE loss will be applied to all spatial areas.

**Super-resolution.** Super-resolution (SR) techniques have emerged as powerful tools for enhancing data resolution, improving the overall quality and fidelity of data representation, and retrieving fine-scale structures. SR is a task that involves recovering fine-scale data from corresponding coarse-grained data. It is also a popular task on PDE learning [9, 68], which is to train SciML models to preserve the inherent physical properties of scientific data.

Motivation: Numerical solutions of PDEs are expected to exhibit invariance to filtering blur or different resolutions of inputs [37, 79]. For instance, in turbulence simulations, the traditional numerical methods always fail to model the expected physical phenomenon with low-resolution meshes due to substantial numerical errors. SR has emerged as a powerful tool for subgrid modeling of PDE dynamics, especially helping to capture the critical patterns of turbulence [51, 37]. Given a specific input distribution, after fitting the SR objective, neural operators are expected to preserve the inherent physical properties and exhibit invariance to filtering blur.

Therefore, we introduce SR as another proxy task. Our objective shares the same motivation as recent SciML works for SR [68]. Specifically, we enforce the model to learn invariant features of unlabeled PDE data that are immune to resolution and blur. Blurry snapshots often occur when the resolution is too low to accurately represent the details in the original content. To do so, we apply a Gaussian filter to blur the unlabeled PDE data, and the autoencoder will reconstruct the high-resolution input with fine-grained details. Instead of applying a fixed blurring, we randomly sample the variance of the Gaussian filter from a certain range as augmentations.

### 3.1.3 PDEs

After pretraining on unlabeled PDE data, we fine-tune neural operators on simulated solutions of PDEs. We study two time-independent PDEs (Poisson, Helmholtz) and two time-dependent PDEs (Reaction-Diffusion, Navier-Stokes). We include details of these PDEs in Appendix A.

### 3.1.4 Model Architectures

We consider two popular architectures for fair comparisons with previous works. These are encoder-decoder architectures designed to reconstruct the original input given partial observations, where the encoder maps observed unlabeled PDE data to a latent space, and the decoder reconstructs the original conditions. We include visualizations of these architectures in Appendix D.

**Fourier Neural Operator.** Fourier Neural Operator (FNO) targets learning PDE data in the Fourier space. The original model backbone (encoder) employs Fourier transform and learns lower Fourier modes with linear transforms. The FNO backbone outputs features back to the spatial domain (i.e., the embeddings are on the pixel level). We refer readers to the original paper for details [45, 46].

- *Pretraining*: We build the decoder to be identical to the encoder (except for the input/output dimension). Unlabeled PDE data are randomly masked at the pixel level.

- *Fine-tuning*: After the pretraining, we discard the decoder, and we follow the original design to append two fully-connected layers (with ReLU activations) to predict final spatial-wise solutions.

**Transformer.** Transformers, which mainly employ self-attention and linear transform blocks, have shown promise in both NLP and CV [78, 13]. Different from FNO, which directly operates on grids, transformers tokenize and group grids into patches, i.e., each tokenized patch embeds a local neighborhood of subgrids. We follow the 3D transformer architecture of Video-MAE [77, 52]. Our encoder embeds patches by a linear projection with added positional embeddings (just as in a standard ViT), and it then processes the resulting set via a series of Transformer blocks. For the transformer, the unlabeled PDE data are randomly masked at the patch level.

- *Pretraining*: For the transformer encoder, we only apply it on the subset of tokens that are visible (i.e., unmasked patches), and masked patches are removed. This allows us to train encoders efficiently. The input to the MAE decoder is the full set of tokens consisting of (i) encoded visible patches, and (ii) the mask token. The mask token is a shared and learned vector that indicates the presence of a missing patch to be predicted. We add positional embeddings to all tokens in this full set. Without this, mask tokens would have no information about their location in the input. Following [28, 77], we adopt an asymmetric design where the decoder is more lightweight (shallower and narrower) than the encoder.

- *Fine-tuning*: After the pretraining, the decoder is preserved during fine-tuning, since we need to reconstruct the tokenized patches back to the input.

### 3.2 Similarity-based Mining of In-Context Examples

Out-of-distribution (OOD) generalization is a critical technical challenge, not only in SciML but also across multiple domains in AI for science [84]. To improve the OOD generalizability of neural operators and to reduce the extra effort of downstream fine-tuning, the following inference paradigm has been proposed: given a query input, the model is also provided with a few supporting examples (dubbed "demos"), together with their ground-truth solutions, to make the final prediction. This approach enables the "open-set" generalization of the model to make predictions on unseen samples.

Originally, in the literature on few-shot learning [82, 83, 55, 70, 35, 48, 61], people developed delicate architectures to find a correspondence between the input and the supporting examples. The purpose of this extra architecture/training design is twofold: first, *find similarities* between the target input and supporting examples; and second, *aggregate labels* of supporting examples for the final predictions. Recent ICL works [80, 81, 47] on learning PDE data also adopt this strategy, with transformers and cross-attention layers.

However, the ICL (in-context learning) of LLMs (large language models) enables a different strategy. The pretraining is still standard and simple (next/masked token prediction), without additional training costs. During inference, LLMs can auto-regressively take any number of few-shot examples, finding similarities between tokens in few-shot examples and those in the target query (via self-attention), and then generate responses by aggregating embeddings of tokens in few-shot examples. This ICL strategy used in LLM is highly scalable and training-efficient.

Motivated by this, we propose to leverage in-context examples via two steps (Algorithm 1).

**Similarity by Prediction.** We find spatial-wise and temporal-wise similar demos by calculating their distance in the output space. That means, for two input locations over the spatial and temporal domains, if we find their outputs of the trained neural operator similar, then we treat them as similar samples. Following [80, 81], we assume demos share the same distribution of physical parameters with the query.

**Aggregation.** For each spatial-temporal location of the query, after finding its similar samples in demos, we aggregate and average their solutions as the prediction.

**Algorithm 1:** Pseudocode of Similarity-based Mining of In-Context Examples.

1 **Data resolution:** x-axis ($W$), y-axis ($H$), output temporal steps ($T$), output channel dimensions for the solution ($C_{out}$).

2 **Input:** Query input ($x$). Paired unlabeled PDE data ($X$) and solutions ($Y \in \mathbb{R}^{J \times H \times W \times T \times C_{out}}$) as $J$ demos. Trained Neural Operator Model $\mathcal{M}$. TopK ($k$) demo solutions to aggregate.

3 $\hat{y} = \mathcal{M}(x)$  ▷ Shape: $H \times W \times T \times C_{out}$

4 $\hat{Y} = \mathcal{M}(X)$  ▷ Shape: $J \times H \times W \times T \times C_{out}$

5 $\hat{\delta} = \hat{y}.\text{reshape}(-1, 1, C_{out}) - \hat{Y}.\text{reshape}(1, -1, C_{out})$  ▷ Query-Demo Distance. Shape: $H \times W \times T \times (J \cdot H \cdot W \cdot T) \times C_{out}$

6 $\hat{\delta} = \text{absolute}(\hat{\delta}).\text{sum}(-1)$

7 index = $\text{argsort}(\hat{\delta}, -1)[:, :, :, : k]$ ▷ Spatial-wise and temporal-wise selection of demos similar to the query.

8 $\hat{y}_{icl} = \text{take\_along\_dim}(Y.\text{reshape}(-1, C_{out}), \text{index})$  ▷ Spatial-wise and temporal-wise aggregation of solutions from similar demos. Shape: $H \times W \times T \times C_{out} \times k$.

9 **Return:** $\hat{y}_{icl}.\text{mean}(-1)$

# 4 Empirical Results

To illustrate the benefits of our approach, we perform empirical evaluations on both PDE benchmarks and real-world observations. *Most importantly*, our unsupervised pretraining (on unlabeled PDE data, followed by fine-tuning) outperforms neural operators trained from scratch, while requiring fewer PDE simulations (Sec. 4.1 and Sec. 4.2). *Moreover*, our in-context examples can help the model generalize better to OOD cases (Sec. 4.3). In our experiments, we trained models three times with different random seeds for statistical significance. For more experimental details, see Appendix B. We also include ablation studies about pretraining hyperparameters in Appendix J. For visualizations of our pretraining, see Appendix K.

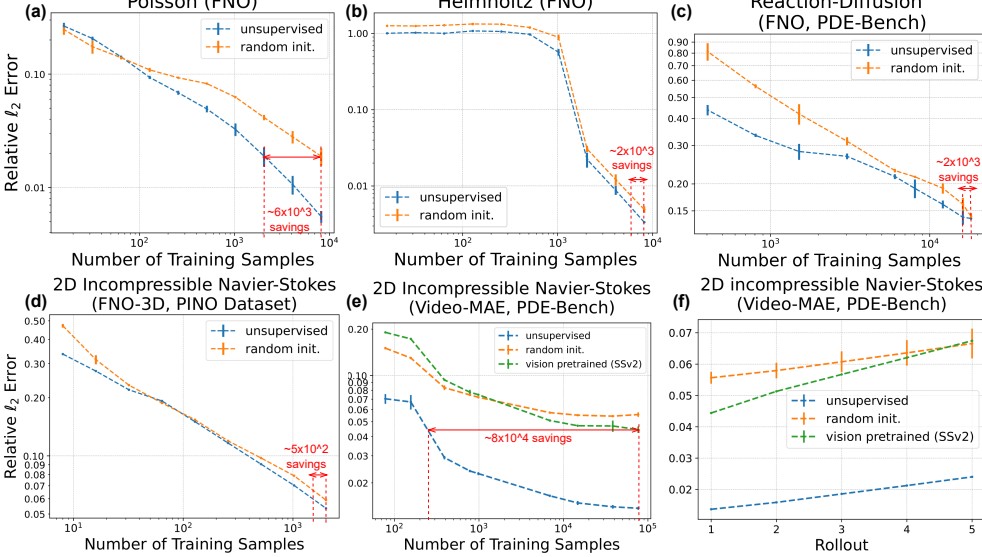

Figure 3: Pretraining neural operators on unlabeled PDE data improves its performance and data efficiency on Poisson (**a**), Helmholtz (**b**), Reaction-Diffusion (**c**), and Navier-Stokes (**d** and **e**, with relative errors at different unrolled steps shown on **f**). "random init.": models are trained from scratch with random initialization. "vision pretrained (SSv2)": fine-tuning from the publicly available checkpoint for Video-MAE (pretrained on computer vision dataset SSV2 [21] for video understanding). Savings of the number of simulated PDE data (when "random init." achieves the best test error) are shown in red.

## 4.1 Unsupervised Pretraining Enables Data-Efficient Operator Learning

**Data Efficiency.**    We first demonstrate that, by leveraging unsupervised pretraining, neural operators can achieve improved errors with less simulated data. In Figure 3, on each PDE, compared with directly training from scratch ("random init."), pretraining on unlabeled PDE data can help neural operators achieve better performance, which can further help reduce the amount of simulated data. Specifically, in our experiments, when we target achieving the best test error of the baseline ("random init."), our method can save $5 \times 10^2 \sim 8 \times 10^5$ simulated solutions across diverse PDE systems.

Among all PDEs, we find that Helmholtz (Figure 3 (b)) is the most challenging. Both two curves failed to improve the error until we increased the number of simulated data points to over 1024. Meanwhile, the generalization gaps remain high (Figure 12 (b)), indicating low training errors. We suspect that learning on the Helmholtz equation may be exhibiting the grokking issue [63, 86], where the network quickly memorizes the training data, but the improvement in generalizability is delayed.

**Unsupervised Pretraining Outperforms Off-the-Shelf Pretrained Checkpoints.**    Pretraining on unlabeled PDE data is not the only way to save simulation costs. As pretraining is widely adopted in CV, pretrained checkpoints on vision data become publicly available and are ready for fine-tuning. We choose to compare with Video-MAE [77] for state-of-the-art video understanding pretrained on SSV2 [21]. As shown in Figure 3 (e), vision-pretrained Video-MAE can only outperform the random initialization with high volumes of simulated data, while its performance suffers when fine-tuned with limited simulations. In contrast, our unsupervised pretraining on unlabeled PDE data can save a significant amount of simulated data.

As errors during testing may quickly accumulate with further timestamps, we also report results with more unrolled steps. We use checkpoints trained with the largest amount of simulated data from above for this study. As shown in Figure 3 (f), at each rollout step, our unsupervised pretraining achieves much better performance. Similarly, vision-pretrained Video-MAE is eventually outperformed by the random initialization at the long rollout step.

**Benefits of Pretraining on Unlabeled PDE Data.**    Pretraining on unlabeled PDE data is beneficial beyond achieving better performance with reduced simulations. *First*, when training on extremely low volumes of data, neural operators tend to overfit, resulting in poor generalization. In Figure 4-left, pretraining on unlabeled PDE data can reduce the generalization gap (testing error − training error). We further show that better generalization gaps persist across all PDEs we studied (see Figure 12). *Second*, pretraining on unlabeled PDE data can lead to faster convergence during fine-tuning. Unlike standard random initializations from Gaussian distributions, pretraining on unlabeled PDE data will provide neural operators with domain-adapted initializations and facilitate a much faster convergence rate, as shown in Figure 4-middle[3]. *Third*, unsupervised pretraining can also help models extract useful representations for subsequent operator learning. From Figure 4-right, we find that even with pre-extracted features (i.e., fixed encoder and only fine-tuned decoder), our neural operators can still outperform the baselines where both the encoder and decoder are updated during fine-tuning.

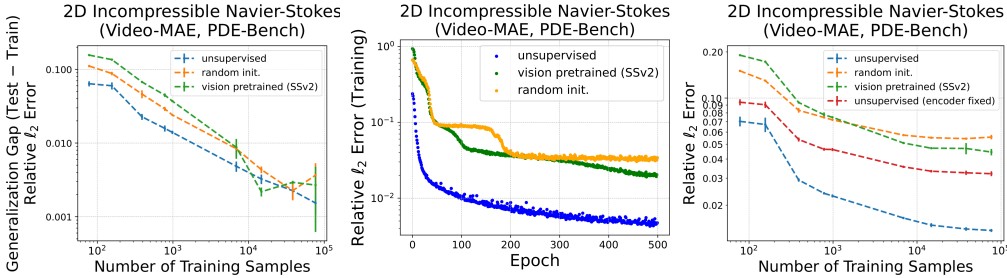

Figure 4: Benefits of our unsupervised pretraining. **Reduced overfitting** (left): our method consistently leads to smaller generalization gaps (test error − training error) across all PDEs we studied (Fig. 12). **Faster convergence** (middle): our unsupervised pretraining can accelerate model convergence than both random initialization and vision-pretrained checkpoint. **Meaningful representations** (right): fine-tuning Video-MAE with fixed encoder (pretrained on unlabeled PDE data, red line) can extract meaningful features and outperform the baseline and the vision-pretrained model (both encoder and decoder are updated during fine-tuning).

---

[3]Training curves when the number of training samples of 14760.

## 4.2 More Comprehensive Experiments on Real-World Data

We now move to a broader range of benchmarks. We will study real-world and noisy data instead of toy datasets, providing even more comprehensive experiments. These benchmarks are widely studied in previous works [60, 62, 52].

**Datasets.**   We brief the background, with more details in Appendix L and visualizations in Fig. 15.

- ECMWF Reanalysis v5 (ERA5) [30] is a public extensive dataset, which delivers hourly data on multiple atmospheric variables spanning from 1979 to today. ERA5 represents a type of atmospheric reanalysis dataset [34], integrating observations from a variety of measurement sources with numerical models through data assimilation [33]. Essentially, it reconstructs the most accurate estimation of the Earth's atmospheric conditions over time. This dataset has been extensively utilized in prior SciML studies [60, 68]. We focus on forecasting the important and challenging *temperature* atmospheric variable.

- ScalarFlow [14] is a reconstruction of real-world smoke plumes. It assembles the first large-scale dataset of realistic turbulent flows. The availability of a large, volumetric data set opens up a wide range of applications, including re-simulations, novel visualization, and metric evaluations. The dataset contains 104 real-world smoke flow density reconstructions. Each reconstruction is captured from five different viewpoints for 150 temporal frames spanning 2.5 seconds.

- Airfoil [75] is a large-scale dataset that contains the pressure and velocity simulations of the flow around real airfoils. This dataset has 53880 samples for training and 90 samples for testing. Each sample contains 6 channels. The 3 input channels include the binary spatial mask of the airfoil and the initial velocity of the freestream condition in the x and y directions. The output channels include the pressure and the x and y velocity components from the simulation at the steady state.

**Model and Training.**   For time-dependent ERA5 (Sec. L.1) and ScalarFlow (Sec. L.2) datasets, we adopt the same VideoMAE architecture [77] used in Sec. 4.1. We use 15 consecutive temporal snapshots to forecast the next time step. We train VideoMAE with Adam, with other hyperparameters the same as in Table 3 column "N.S. (PDEBench)". For the time-independent steady-state Airfoil dataset (Sec. L.3), we adopt the 2D-FNO architecture. We train 2D-FNO with Adam, with other hyperparameters the same as in Table 3 column "Poisson".

**Results.**   As shown in Figure 5, compared with directly training operators from scratch ("random init."), pretraining on unlabeled data (2D snapshots of ERA5/ScalarFlow without temporal dynamics, or freestream velocities of Airfoil) can help neural operators achieve better performance on both temperature/flow forecasting and predictions of the steady-state pressure and velocity around airfoils.

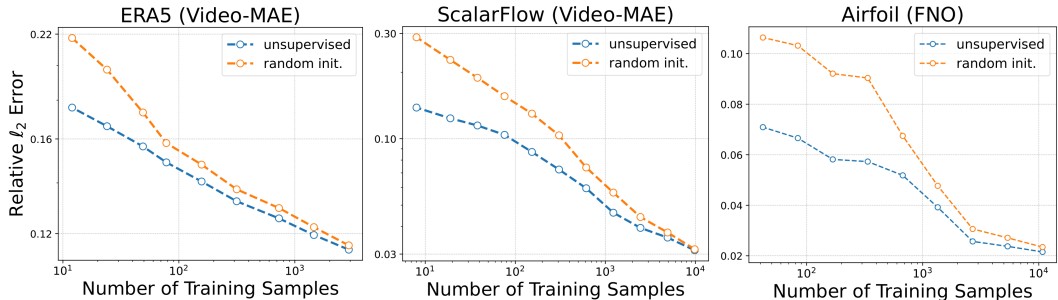

Figure 5: For **real-world scientific problems**, pretraining neural operators on unlabeled PDE data improves its performance and data efficiency. We study VideoMAE [77] pretrained with unlabeled snapshots (no temporal dynamics), and then fine-tune across different numbers of temporal snapshots on ERA5 (left) and ScalarFlow (middle). We also pretrain 2D-FNO [45] on freestream velocities and fine-tune on time-independent steady-state airflow pressure and velocities (right). "random init.": models are trained from scratch with random initialization.

## 4.3 In-Context Examples Enable Data-Efficient OOD Generalization

We now move to OOD settings, where models will be tested on PDE data simulated with physical parameters unseen during fine-tuning/training. We include how to simulate OOD samples in

Appendix B.2. Neural operators suffer from poor OOD generalization [73, 80]. Traditionally, improvements heavily depend on further fine-tuning on simulated data, which requires extra simulation and training costs. We study the benefits of leveraging test-time in-context examples. As shown in Figure 6, when we flexibly scale up the number of demos, we can keep improving FNO's OOD generalization on diverse PDEs. We follow [80, 81] that demos are randomly sampled from the same distribution used to generate the OOD test set. When the number of demos is 0, we have the baseline in the OOD setting. Notably, we introduce zero training overhead: we keep the standard training pipeline, and our mining of in-context examples can be seamlessly plugged in during OOD inference.

We further provide a baseline, which uses features extracted by the backbone of the neural operator (high-dimensional features before the final output layer) to find similar samples. As we can see, this baseline is worse than our method (both performance and confidence), indicating that the final output of the neural operator can more accurately indicate true similar samples.

See Appendix M for further discussions about the benefits of leveraging in-context examples, and Appendix N for visualizations.

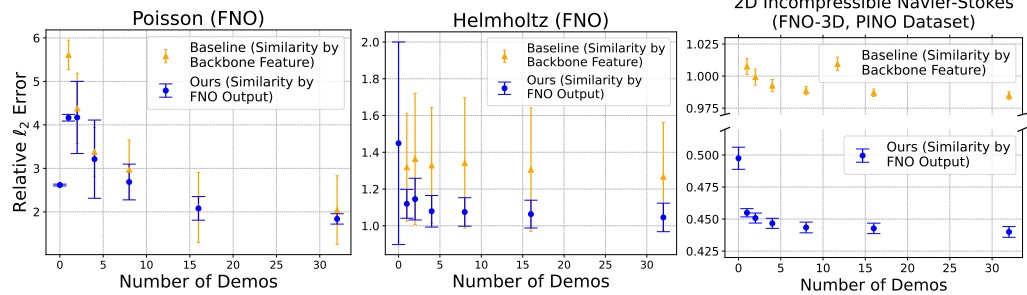

Figure 6: In-context examples for OOD testing. **Our method (blue) improves both $\ell_2$ errors and confidence as we increase the number of demos.** "Ours (Similarity by FNO Output)": we leverage the output (prediction) of neural operators to find similar samples. "Baseline (Similarity by Backbone Feature)": the baseline uses features extracted by the backbone of the neural operator (high-dimensional features before the final output layer) to find similar samples.

## 5    Conclusion

In this work, we focus on improving the data efficiency of solving partial differential equations (PDEs) using deep learning, with a particular emphasis on unsupervised pretraining and in-context learning (ICL) methods. Our key contributions include introducing unsupervised pretraining for operator learning and a flexible ICL approach that enhances out-of-distribution (OOD) generalization without increasing training costs. Through extensive evaluations, we demonstrate that our method is not only more data-efficient, but it also achieves greater generalizability compared to existing approaches. By improving the data efficiency of neural operators for solving PDEs, our approach can significantly reduce the computational costs and energy demands of high-fidelity numerical PDE simulations. Additionally, by making advanced PDE solutions more accessible through efficient pretraining, our method has the potential to accelerate scientific and engineering progress across various fields, ultimately benefiting society. We hope our work will inspire the scientific machine learning (SciML) community to further address the high simulation costs and limited OOD generalization of neural operators, contributing to advancements that support both scientific innovation and environmental sustainability.

## 6    Limitations

Current limitations of our work: 1) We could design more physics-inspired proxy tasks and data augmentation methods for scientific data; 2) We could study more PDE systems in our unsupervised pretraining and in-context learning; 3) We could consider more different neural operator architectures. We expect that addressing these limitations will lead to broader impacts in future works.

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

# A PDEs

We consider two time-independent PDEs (Poisson, Helmholtz) and two time-dependent PDEs (Reaction-Diffusion, Navier-Stokes), as described below. We also describe types of unlabeled data for per-PDE examples. For a general definition of unlabeled PDE data, please refer to Section 3.1.1.

1. *Poisson*: We consider a two-dimensional (2D) elliptic PDE that arises in various physical situations, with periodic boundary conditions within a spatial domain $\Omega = [0, 1]^2$:

$$-\text{div}\, \boldsymbol{K}\nabla u = f \quad , \tag{1}$$

   where $u(x)$ represents the solution, $f(x)$ acts as the source (e.g., forcing) function, and $x$ denotes spatial coordinate. The diffusion coefficient tensor, denoted by $\boldsymbol{K}$, is employed to measure the physical properties of this system.

   **Unlabeled PDE Data**: In [73], inputs to neural operators have four channels: the source function, and three diffusion coefficients (two diagonal elements and one off-diagonal element in the symmetric diffusion coefficient tensor) expanded to the whole spatial domain. We use this input as the unlabeled data to pretrain the neural operator.

2. *Helmholtz*: We consider the 2D inhomogeneous Helmholtz equation, which is a time-independent form of the wave equation, with periodic boundary conditions and a spatial domain $\Omega = [0, 1]^2$. The governing equation is given by

$$-\Delta u + \omega u = f \quad \text{in } \Omega, \tag{2}$$

   where $u(x)$ denotes the solution, $f(x)$ represents the source function, and $\omega > 0$ is the wavenumber (that defines the dynamic properties of the Helmholtz equation). This system produces high-frequency large wavenumber oscillatory patterns, which poses challenges in terms of generalization.

   **Unlabeled PDE Data**: In [73], inputs to neural operators have two channels: the source function, and the wavenumber $\omega$ expanded to the whole spatial domain. We use this input as the unlabeled data to pretrain the neural operator.

3. *Reaction-Diffusion:* The 2D Reaction-Diffusion (RD) equation involves the interaction between two nonlinear coupled variables, i.e., the activator $u(t, x, y)$ and the inhibitor $v(t, x, y)$. The RD equation is given by

$$\begin{aligned} \partial_t u &= D_u(\partial_{xx} u + \partial_{yy} u) + R_u, \\ \partial_t v &= D_v(\partial_{xx} v + \partial_{yy} v) + R_v, \end{aligned} \tag{3}$$

   where $\{D_u, D_v\}$ are diffusion coefficients for $u$ and $v$, respectively, and where $R_u$ and $R_v$ are reaction functions, which are defined as the Fitzhugh-Nagumo equation [36]:

$$\begin{aligned} R_u &= u - u^3 - k - v, \\ R_v &= u - v. \end{aligned} \tag{4}$$

   The spatial domain considered for the 2D RD equation is $\Omega = [-1, 1]^2$, and the time duration is $t \in (0, 5]$.

   **Unlabeled PDE Data**: In PDEBench [74], we forecast the activator $u$ and the inhibitor $v$ (i.e. two input channels) with $T = 10$. Since each individual snapshot can serve as an initial condition for forecasting, during our unsupervised pretraining we discard the temporal dynamics and use randomly shuffled snapshots of $u$ and $v$ with $T = 1$ as unlabeled PDE data. That means, during pretraining the model only has access to single and individual snapshots without any awareness of temporal dynamics.

4. *Navier-Stokes Equation:* Lastly, we consider the 2D incompressible Navier-Stokes equation in vorticity form on the unit torus, which is formulated as

$$\begin{aligned} \nabla \cdot \mathbf{v} &= 0 \\ \rho\left(\partial_t \mathbf{v} + \mathbf{v} \cdot \nabla \mathbf{v}\right) &= -\nabla p + \nu \nabla^2 \mathbf{v} + \mathbf{f} \end{aligned} \tag{5}$$

   Here, the velocity $\mathbf{v}$ is defined within the time duration $[0, T]$ and the spatial domain $[0, 1]^2$ (the vorticity can be formulated as $w = \nabla \times \mathbf{v}$). Moreover, $\rho$ is the density, and the coefficient

$\nu$ signifies the viscosity of a fluid, and $\mathbf{f}$ is the external forcing function. Dirichlet boundary conditions are employed in this system.

**Unlabeled PDE Data**: We follow the training settings from previous works. 1) For FNO [45], we focus on mapping the vorticity from the initial condition to the full solution ($w_0 \mapsto w|_{[0,T)}$, with $T = 33$). The input includes four channels: the vorticity (which is the initial condition, duplicated for $T$ times), and three (xy-spatial and temporal) linear mesh position embedding channels. Thus, similar to Reaction-Diffusion above, during pretraining, the model will take individual snapshots of vorticity and position embeddings as unlabeled data. 2) For PDEBench [74], we forecast both xy velocities and pressure (i.e. three input channels), with $T = 15$. The model will take individual snapshots of vorticity and xy velocities as unlabeled data.

We summarize detailed inputs and outputs in Table 1.

Table 1: Inputs and outputs for learning different PDEs. See Table 3 for resolutions. "NS": Navier-Stokes. "RD": Reaction-Diffusion.

| PDE Simulations | Input | Input Shape | Output |
|---|---|---|---|
| Poisson [73] | Source function ($f$), diffusion coefficients ($\boldsymbol{K}$) | $C \times H \times W(C = 4)$ | Potential field ($u$) |
| Helmholtz [73] | Source function ($f$), wavenumber $\omega$ | $C \times H \times W(C = 2)$ | Wave function ($u$) |
| NS (FNO [45]) | Vorticity ($w$), Spatiotemporal Coordinates | $H \times W \times T \times 4(T = 33)$ | Vorticity ($w \in [0, T)$) |
| NS (PDEBench [74]) | Velocity ($v_x, v_y$), pressure ($p$) | $T \times C \times H \times W(T = 15, C = 3)$ | Velocity ($v_x, v_y$), pressure ($p$) at $T + 1$ |
| RD (PDEBench [74]) | Activator ($u$), inhibitor ($v$) | $T \times C \times H \times W(T = 10, C = 2)$ | Activator ($u$), inhibitor ($v$) at $T + 1$ |

# B  Detailed Experiment Settings

## B.1  Distributions of Unlabeled PDE Data

In Table 2, for the purpose of OOD testing, we summarize the distribution of our unlabeled PDE data during pretraining, fine-tuning, and inference with in-context examples. Ranges of these physical parameters are inspired by [73]. During pretraining, we consider a wide distribution of unlabeled PDE data. When training (fine-tuning) our model, we consider in-distribution unlabeled PDE data. Finally, we test our similarity-based method that learns in-context examples in OOD settings.[4] For Helmholtz OOD, we choose a narrow range of coefficients ([15, 20]) mainly because its solution is very sensitive to the wavenumber, and FNO's performance significantly drops when we move to more extreme OOD settings.

Table 2: Ranges of physical parameters (integers) for unsupervised pretraining, training (fine-tuning), and out-of-distribution (OOD) inference.

| Physics Parameters | Poisson (diffusion) | Helmholtz (wave number) | Naiver Stokes (Reynolds number) |
|---|---|---|---|
| Unsupervised Pretraining | [1, 20] | [1, 20] | {100, 300, 500, 800, 1000} |
| Training (or Fine-tuning) | [5, 15] | [5, 15] | 300 |
| Out-of-Distribution Testing | [15, 50] | [15, 20] | 10000 |

## B.2  Data Generation

**Unlabeled PDE Data.**  We generate data for Poisson and Helmholtz [73], Reaction-Diffusion on PDE-Bench [74] and 2D incompressible Navier-Stokes on PINO Dataset [46] following the procedure mentioned in the paper. For unlabeled data generation, we bypass the computation of solvers, which expedites the generation speed, as shown in Table 4.

**OOD Samples.**  The OOD data generation procedure is similar to the unlabeled data, except for the changes in the physical parameters coefficients. For Poisson and Helmholtz, we consider changing the range of diffusion eigenvalue and waver number respectively. For Navier-Stokes equation, we change the Reynolds number. We list the coefficients in Table 2.

---

[4]For Navier Stokes from PDEBench, we use the original data. We could not generate our own pretraining/finetuning data with different ranges of physics parameters, due to a possible mismatch of the provided configuration files and the version of Phiflow used in PDEBench (see GitHub issue at https://github.com/pdebench/PDEBench/issues/36).

## B.3 Training Hyperparameters

We summarize our hyperparameters used during pretraining and fine-tuning/training in Table 3. These hyperparameters strictly follow previous works [45, 73, 74, 46, 52]. We conducted our experiments on four A100 GPUs, each with 40GB of memory.

Table 3: Hyperparameters for pretraining and training/fine-tuning. "N.S.": 2D Incompressible Navier-Stokes. "DAdapt": adaptive learning rate by D-adaptation [10]. "ns": total number of simulated training samples. A batch size of "$\min(32, \text{ns})$" is because the total number of training samples might be fewer than 32.

| Stage ↓ | PDEs → | Poisson | Helmholtz | Reaction-Diffusion | N.S. (PINO) | N.S. (PDEBench) | ERA5 | ScalarFlow | Airfoil |
|---|---|---|---|---|---|---|---|---|---|
| Pretraining | Number of Samples | 46,080 | 46,080 | 76,760 | 57,545 | 713,286 | 8,760 | 56,160 | 43,104 |
| | Learning Rate | $1 \times 10^{-3}$ | $1 \times 10^{-3}$ | $1 \times 10^{-3}$ | $1 \times 10^{-3}$ | DAdapt | $1 \times 10^{-3}$ | $1 \times 10^{-3}$ | $1 \times 10^{-3}$ |
| | Batch Size | 32 | 32 | 5 | 2 | 8 | 32 | 32 | 32 |
| | Resolution: ($H \times W$) | 64×64 | 64×64 | 128×128 | 128×128 | 512×512 | 180×180 | 180×120 | 96×45 |
| | Epochs/Iterations | 500 epochs | 500 epochs | 500 epochs | 100,000 iters | 500 epochs | 500 epochs | 500 epochs | 500 epochs |
| Training / Fine-tuning | Learning Rate | $1 \times 10^{-3}$ | $1 \times 10^{-3}$ | $1 \times 10^{-3}$ | $1 \times 10^{-3}$ | DAdapt | $1 \times 10^{-3}$ | $1 \times 10^{-3}$ | $1 \times 10^{-3}$ |
| | Batch Size | min(32, ns) | min(32, ns) | 5 | 2 | 8 | 4 | 4 | min(32, ns) |
| | Resolution: ($H \times W$ or $H \times W \times T$) | 64×64 | 64×64 | 128×128×10 | 64×64×33 | 512×512×15 | 180×180×16 | 180×120×16 | 96×45 |
| | Epochs/Iterations | 500 epochs | 500 epochs | 500 epochs | 50,000 iters | 500 epochs | 500 epochs | 500 epochs | 500 epochs |
| | Rollouts | N/A | N/A | 91 | N/A | 1 | 1 | 1 | N/A |

## C  Examples of Simulation Costs

In Table 4, we demonstrate the cheap simulation of only unlabeled PDE data, versus simulating both unlabeled PDE data and solutions, on 2D incompressible Navier-Stoke on PINO Dataset [46] and Reaction-Diffusion on PDE-Bench [74]. We can see that unlabeled PDE data are extremely cheap to simulate. Therefore, our pretraining method can boost the performance and meanwhile save the heavy cost of data simulations.

Table 4: Simulation time costs on 2D Incompressible Navier-Stokes ("N.S.") on PINO Dataset [46] and Reaction-Diffusion ("R.D.") on PDE-Bench [74]. "$Re$": Reynolds number. "$D_u, D_v$": diffusion coefficients. $N$: number of samples. $T$: temporal resolution. $H \times W$: spatial resolution. $C$: input channels (1 for the vorticity in N.S., 2 for velocities $u, v$ in R.D.).

| Data | Physical Parameters | Unlabeled PDE Data (sec.) | Data+Solutions (sec.) | Data Size | CPU | GPU |
|---|---|---|---|---|---|---|
| N.S. | $Re = 100$ | 5499.32 | 11013.90 | | | |
| | $Re = 300$ | 3683.02 | 7625.82 | $N \times T \times H \times W =$ | 1 AMD EPYC 7763 | 1 NVIDIA A100 (40GB) |
| | $Re = 500$ | 4059.71 | 8963.39 | $20 \times 2000 \times 512 \times 512$ | | |
| | $Re = 800$ | 4829.3 | 10811.15 | | | |
| | $Re = 1000$ | 4957.24 | 10788.69 | | | |
| R.D. | $D_u = 1 \times 10^{-3}$ $D_v = 5 \times 10^{-3}$ | 29.65 | 6657.34 | $N \times T \times H \times W \times C =$ $1000 \times 101 \times 128 \times 128 \times 2$ | 1 AMD EPYC 7763 | N/A |

## D  Model Architectures

In Figure 7, We show visualizations of architectures described in Sec. 3.1.4. Specifically, the FNO has 67.1M parameters, and the Video-MAE has 23.4M. During pretraining, FNO costs 20 GPU hours, and Video-MAE costs 18 GPU hours. During fine-tuning, FNO costs 4 GPU hours, and Video-MAE costs 6 GPU hours.

## E  Comparison with Contrastive Learning

Contrastive learning is an important self-supervised pretraining technique studied in computer vision. For a fair comparison, we directly compare with MoCo v2 [8], a highly-cited self-supervised learning method also originally implemented in PyTorch, whose core method is closely related to SimCLR [7] (originally implemented in TensorFlow).

We compare MoCo v2 with our method on a broader real-world benchmark ERA5 [30]. As shown in Figure 8, our unsupervised learning method can largely outperform MoCo v2. This extra comprehen-

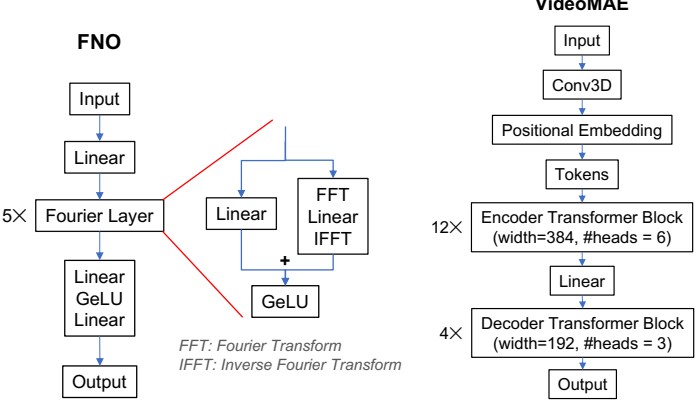

Figure 7: Visualizations of architectures we studied. Left: FNO [46]. Right: VideoMAE [77].

sive result demonstrates that our method can be widely adopted in real-world problems, outperforming previous unsupervised learning methods.

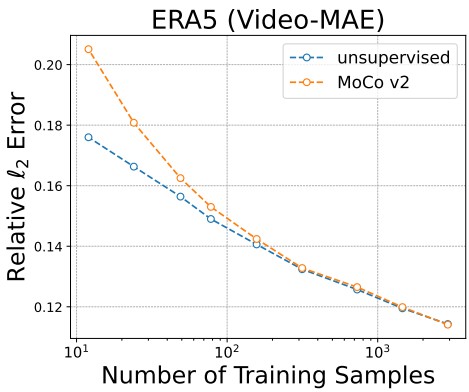

Figure 8: Comparison between our unsupervised pretraining method versus MoCo v2 [8].

## F   More Comparisons with Vision Pretrained Models

Beyond Figure 3(e) for the transformer on time-dependent PDE (Navier-Stokes), we further verify that pretraining on unsupervised PDE data makes FNO outperform its off-the-shelf vision-pretrained checkpoints. Specifically, we first pretrain the 2D FNO model on ImageNet [11], then fine-tune on the downstream PDE simulation data. As shown in Figure 9, vision-pretrained FNO performs worse during downstream fine-tuning on time-independent PDEs (including both Poisson and Helmholtz equations), confirming that domain-specific pretraining, even on unsupervised PDE data, is more beneficial than conventional checkpoints pretrained on unrelated domains like computer vision.

## G   Joint Pretraining Further Improves Performance

We also study if joint pretraining on unlabeled multiple PDE data can bring extra benefits. We combine all 46,080 unlabeled Poisson samples and all 46,080 unlabeled Helmholtz samples (see Table 3). We choose this setting because recent works on SciML foundation models [52, 26] also use all samples from each PDE for pretraining. We do zero-paddings for mismatched channels. From Figure 10, we can see that joint pretraining can further improve the performance of fine-tuning on different PDEs.

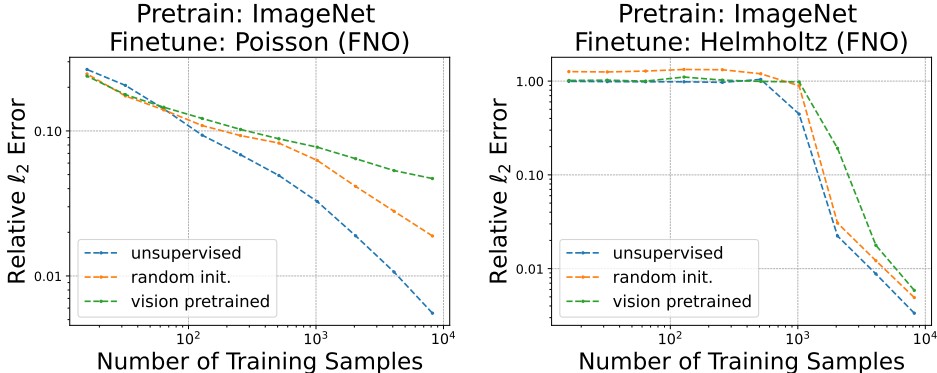

Figure 9: Pretraining neural operators on unlabeled PDE data improves its performance and data efficiency on Poisson (left), Helmholtz (right). "random init.": models are trained from scratch with random initialization. "vision pretrained": fine-tuning from the checkpoint pretrained on computer vision dataset ImageNet [11].

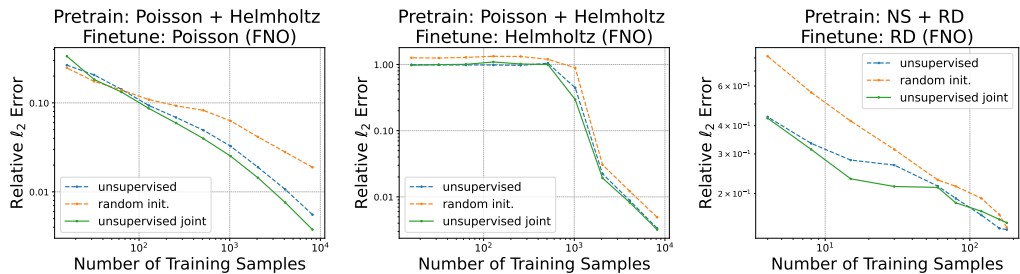

Figure 10: **Joint unsupervised pretraining on multiple PDEs (green solid curve) further improves the data efficiency of neural operators** when fine-tuning on Poisson (left), Helmholtz (middle), Reaction-Diffusion (right). "random init.": models are trained from scratch with random initialization. "unsupervised": models are pretrained on a *single* unsupervised PDE data. "unsupervised joint": models are pretrained on a *joint* of multiple unsupervised PDE datasets. "NS": Navier Stokes. "RD": Reaction-Diffusion.

## H Fine-tuning on Unseen PDEs is Challenging

We also try to fine-tune neural operators on unseen PDEs (i.e. PDEs different from pretraining). Mismatched channels are padded with zeros. We find this will lead to worse performance compared with models pretrained on the same PDE, as shown in Figure 11.

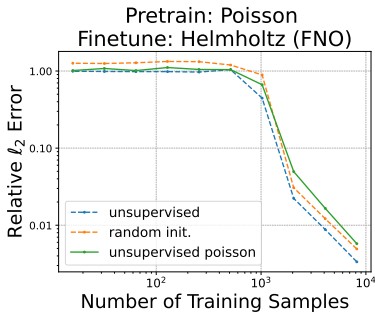

Figure 11: Fine-tuning FNO (pretrained on Poisson) on unseen samples from Helmholtz.

## I Consistently Improved Generalization

Beyond the generation gap we have shown in Figure 4 (left) for the Navier Stokes equation from PDEBench, we further collect generation gaps of our models learned on other PDEs. As shown in Figure 12, on diverse PDE systems, our method can contribute to universally reduced overfitting.

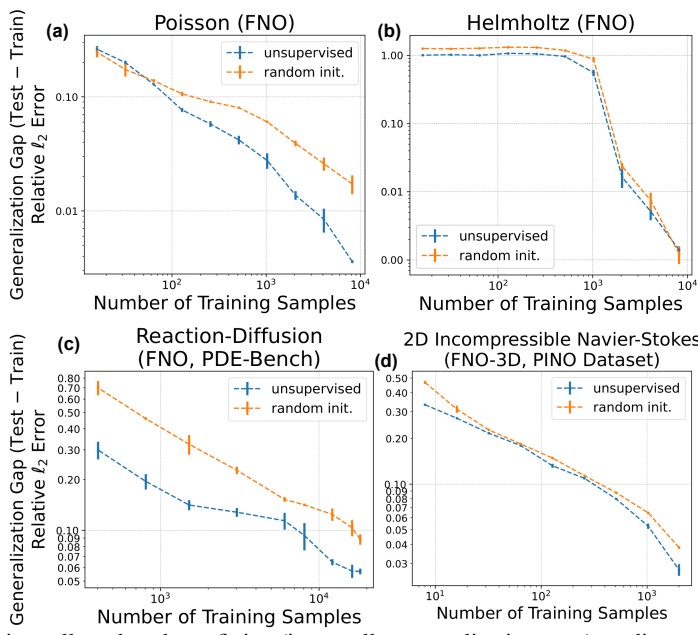

Figure 12: Universally reduced overfitting (i.e. smaller generalization gaps) on diverse PDEs (**a** to **d**).

## J  More Ablation Studies

### J.1  Magnitude of Perturbations during Pretraining.

We study the optimal magnitude of perturbations during pretraining: "Mask Ratio" (1 indicates no visible grids, and 0 means no masks); and "Blur Sigma" for the variance of Gaussian kernel for blur (larger the more degradation). We show our results in Table 5, 6, 7, 8. For example, on 2D incompressible Navier-Stokes with FNO, as shown in Table 8, we can find that when training with a low volume of data, we should use much stronger perturbations (high masking ratios and strong blur), whereas a high volume of data only requires mild perturbations.

Table 5: Best choice of mask ratio and blur sigma for pretraining on Poisson equation.

| #Samples | Mask Ratio | Blur Sigma |
|---|---|---|
| 16 | 0 | 0 |
| 32 | 0 | 0∼1 |
| 64 | 0 | 0∼1 |
| 128 | 0 | 0∼1 |
| 256 | 0 | 0∼1 |
| 512 | 0 | 0∼1 |
| 1024 | 0 | 0∼1 |
| 2048 | 0 | 0∼1 |
| 4096 | 0 | 0∼1 |
| 8192 | 0 | 0∼1 |

Table 6: Best choice of mask ratio and blur sigma for pretraining on Helmholtz equation.

| #Samples | Mask Ratio | Blur Sigma |
|---|---|---|
| 16 | 0.2 | 0∼1 |
| 32 | 0.2 | 0∼1 |
| 64 | 0.2 | 0∼1 |
| 128 | 0.2 | 0∼1 |
| 256 | 0.2 | 0∼1 |
| 512 | 0.6 | 0∼2 |
| 1024 | 0.6 | 0∼2 |
| 2048 | 0 | 0∼1 |
| 4096 | 0 | 0∼1 |
| 8192 | 0 | 0∼0.5 |

### J.2  Ablation of the Number of Pretraining Samples.

As shown in Table 9, the more unlabeled PDE data we use for pretraining, the better quality the pretrained model will be.

Table 7: Best choice of mask ratio and blur sigma for pretraining on 2D Diffusion-Reaction equation.

| #Samples | Mask Ratio | Blur Sigma |
|---|---|---|
| 404 | 0 | 0∼1 |
| 808 | 0 | 0∼1 |
| 1515 | 0 | 0∼1 |
| 3030 | 0.7 | 0∼2 |
| 6060 | 0.9 | 0∼1 |
| 8080 | 0.3 | 0∼1 |
| 12120 | 0 | 0∼1 |
| 16160 | 0 | 0∼1 |
| 18180 | 0 | 0∼1 |

Table 8: Best choice of mask ratio and blur sigma for pretraining on 2D incompressible Navier-Stokes.

| #Samples | Mask Ratio | Blur Sigma |
|---|---|---|
| 8 | 0.7 | 0∼4 |
| 16 | 0.7 | 0∼4 |
| 32 | 0.7 | 0∼4 |
| 64 | 0 | 0∼1 |
| 128 | 0 | 0∼1 |
| 256 | 0.05 | 0 |
| 512 | 0.05 | 0 |
| 1024 | 0.05 | 0 |
| 2000 | 0.05 | 0 |

Table 9: More unlabeled PDE data improve the quality of pretraining. FNO on 2D incompressible Navier-Stokes, pretrained with mask ratio as 0.7.

| #Pretraining Samples | 2000 | 57545 |
|---|---|---|
| Relative $\ell_2$ Error | 0.3594 | 0. 3246 |

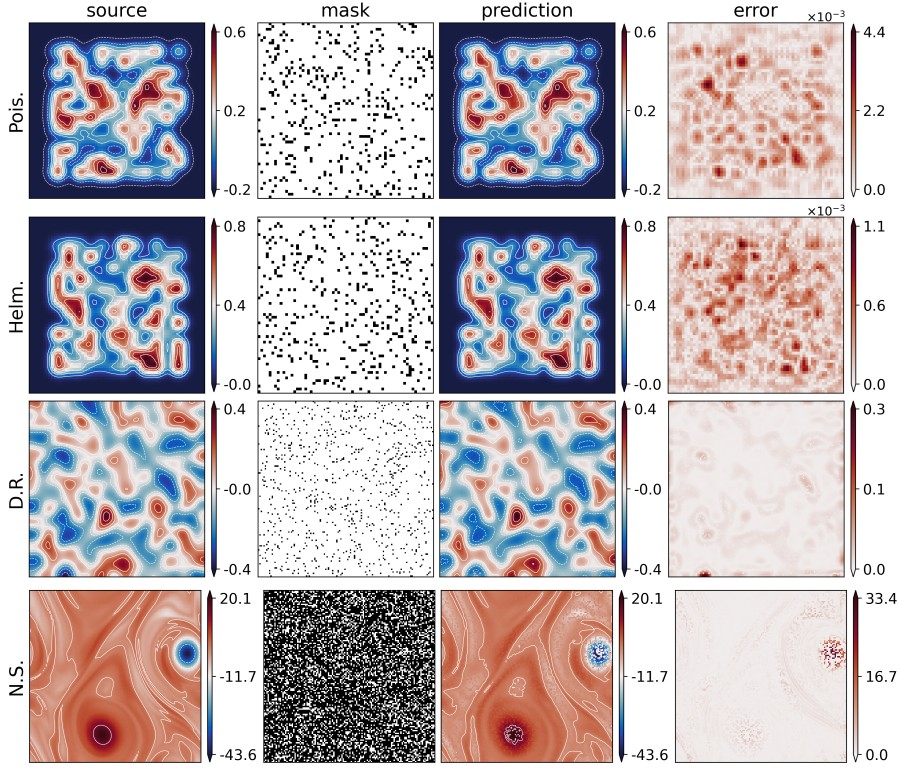

Figure 13: Visualization of FNO reconstructions of unlabeled PDE data on the Poisson ("Pois."), Helmholtz ("Helm."), 2D Diffusion-Reaction ("D.R."), and 2D incompressible Navier-Stokes ("N.S.") equations during MAE pretraining. (Mask ratio: 0.1 for Poisson, Helmholtz, and 2D Diffusion-Reaction equations; 0.7 for incompressible Navier-Stokes.) In masks, only white areas are visible to the model during pretraining.

# K  Visualization of MAE Pretraining

To demonstrate the efficacy of our MAE-based pretraining, we show the unlabeled PDE data and its reconstructed version in Figure 13 (MAE pretraining on different PDEs) and Figure 14 (MAE pretraining on 2D incompressible Navier-Stokes with varying mask ratios). We can see that all inputs are accurately reconstructed with low errors and similar patterns.

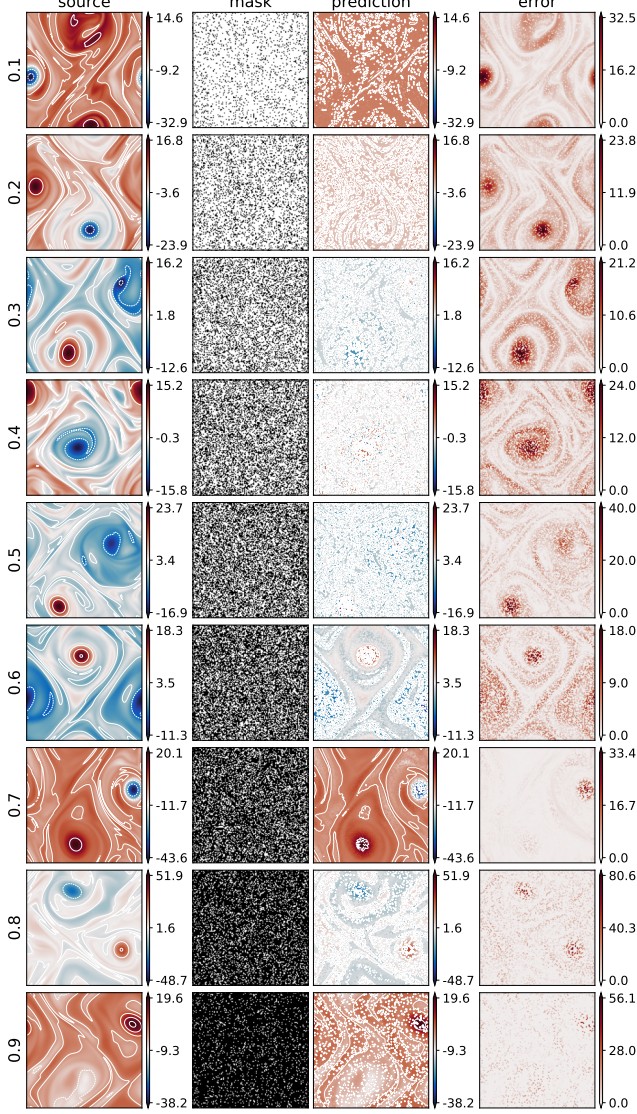

Figure 14: Visualization of FNO reconstructions of unlabeled PDE data on the 2D incompressible Navier-Stokes equations during MAE pertaining with mask ratio from 0.1 to 0.9.

# L  Details and Visualizations of Real-World Data

## L.1  ECMWF Reanalysis v5 (ERA5)

We utilize data from 2006 to 2015, with the snapshots taken every 6 hours and a spatial resolution of $360 \times 360$. The total number of snapshots is 14600. We apply the mean-standard deviation normalization to the data and downsample the snapshots to a spatial resolution of $180 \times 180$. We split 75% of the data for pretrain and 25% for finetune. For each split, we further separate 80% of the data for training, 10% for validation, and 10% for testing.

## L.2 ScalarFlow

The original spatial resolution is $1062 \times 600$. We crop the snapshots to $900 \times 600$ to remove the padding and background. We remove the first 15 timeframes for each simulation to avoid the initial transient phase at the beginning of the smoke flow generation. We then downscale the snapshots to $180 \times 120$ and apply mean-standard deviation normalization to the data. With a total of 70200 snapshots, we split 80% of the data for pretrain and 20% for finetune. For each split, we further separate 80% of the data for training, 10% for validation, and 10% for testing.

## L.3 Airfoil

We split 80% of the training data for pretrain and 20% for finetune. The original spatial resolution of each sample is $128 \times 128$. We crop the snapshots to $96 \times 45$ to remove the background. We then apply min-max normalization channel-wise to the sample.

We show visualizations of real-world scientific data in Figure 15.

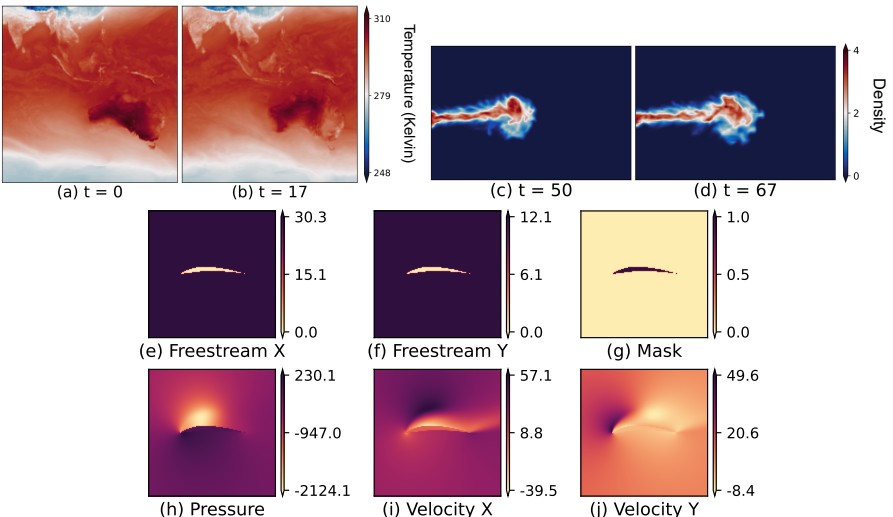

Figure 15: We show snapshot examples from ERA5 temperature [30] (**a**, **b**) and ScalarFlow [14] (**c**, **d**) at different temporal steps; and also an example of Airfoil mask, velocities, and pressure [75] (**e-j**).

## M Benefits of In-context Examples

We try to understand the benefit of leveraging in-context examples by decomposing the relative MSE error into "scale" and "shape." "Scale" is the slope of a linear regression between targets and predictions (closer to 1 the better), indicating the alignment of the range of model outputs with targets. "Shape" is the normalized relative MSE (i.e., model outputs or targets are normalized by their own largest magnitude before MSE), indicating the alignment of scale-invariant spatial/temporal structures. We show results in Figure 16. We find that the benefit of in-context examples lies in that the scale of the model's output keeps being better calibrated ("scale" being closer to 1) when adding more demos.

In numerical simulations or predictions of PDEs, there are settings where the scale or magnitude of solutions is more important than the exact shape/pattern:

1. Heat Transfer: In large-scale systems, the focus might be on overall temperature and extreme values. For instance, in evaluating a cooling system, the key concern might be the peak temperature rather than the detailed temperature distribution.
2. Fluid Dynamics: For applications like aerodynamics, the overall drag or lift force on an object is often more critical than capturing every detail of the flow pattern, such as in airfoil design.
3. Environmental Modeling: The concentration of pollutants at specific locations or total pollutant transport is often more crucial than the exact distribution, such as in groundwater flow studies.

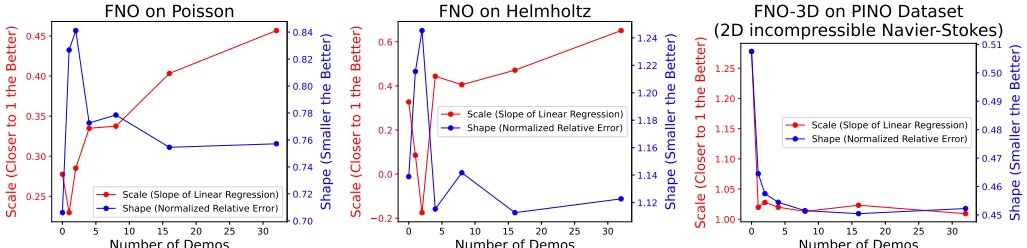

Figure 16: Benefits of in-context examples. To analyze the benefit of in-context examples for complicated PDE systems, we decompose the relative MSE error into "Scale" and "Shape". "Scale" indicates the alignment of the range of model outputs with targets (closer to 1 the better), via the slope of a linear regression. "Shape" indicates the alignment of scale-invariant spatial/temporal structures via normalized relative MSE (i.e. model outputs or targets are normalized by their own largest magnitude before MSE). We find that the benefit of in-context examples lies in that the scale of the model's output keeps being calibrated (red line being closer to 1) when adding more demos.

# N    Visualizations with In-Context Examples

We show visualizations of our similarity-based mining of in-context examples in Figure 17. In this visualization, we find that the range of numerical solutions (e.g., values in colorbars) predicted with in-context examples becomes closer to the target. Meanwhile, based on this visualization, we conclude that OOD generalization is challenging for neural operators because of: 1) Significantly different patterns of solutions under different physical parameters; 2) Different value ranges of solutions under different physical parameters.

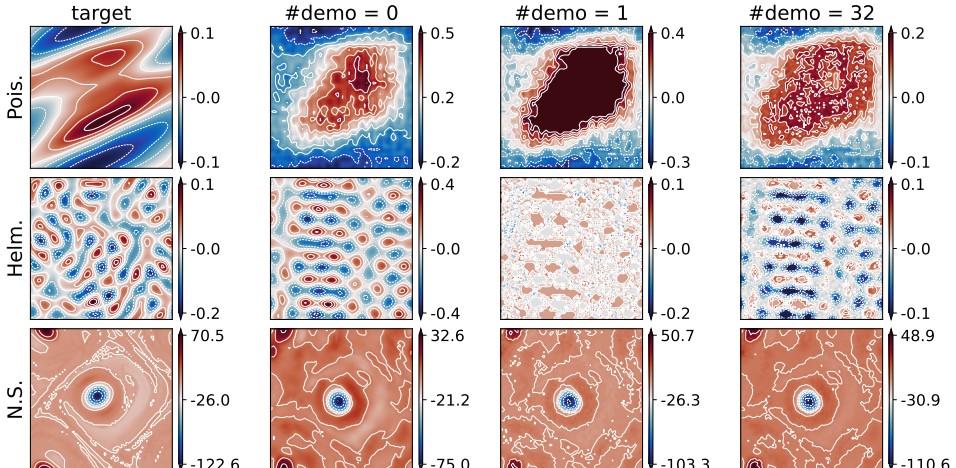

Figure 17: Visualizations of mining in-context examples for FNO in OOD testing. Ranges of solutions predicted with in-context examples (min/max of each snapshot, reflected in colorbars) become closer to the target.

