# OpenReview forum: "Data-Efficient Operator Learning via Unsupervised Pretraining and In-Context Learning"
_NeurIPS.cc/2024/Conference — NeurIPS 2024 poster_

### Official Review · Reviewer_u3i6 · 2024-07-02

**Soundness:** 2
**Presentation:** 3
**Contribution:** 3
**Rating:** 6
**Confidence:** 4

**Summary:**

This paper considers the problem of training neural operators to solve forward partial differential equation (PDE) problems in data-limited settings. The paper motivates this setting with the large data simulation cost incurred by these methods when generating a large training set of PDE solutions. The paper suggests a self-supervised pretraining method derived from self-supervised learning tasks popular in the computer vision community. This pretraining method only requires a dataset of initial conditions, PDE coefficients, and physical parameters; it does not require any PDE solutions. The paper shows the benefit of their method by showing their pretrained models outperform randomly-initialized models or models pretrained on general computer vision tasks; there are many evaluations across a range of PDE problems and real-world experimental data. In addition, the paper considers an "out-of-distribution" (OOD) task where PDE coefficient settings are changed at test time and suggests an "In-Context Learning" (ICL) strategy, which uses the trained model to adaptively combine examples of ground-truth solutions.

**Strengths:**

This paper addresses important challenges in the field of neural operators and provides well-motivated solutions to these problems. There are a large number of experiments to quantify the effect of these solutions. I will list individual strengths below:
- The paper considers important problems in the field of neural operators: reducing the dependence on training data generated by PDE solvers, adapting trained models to out-of-distribution problem settings, and quantifying performance on complicated real-world datasets.
- The experimentation in this paper is very thorough. The paper shows experiments on PDE problems across a broad range of physical parameters as well as real-world datasets ERA5 temperature, ScalarFlow, and Airfoil.
- The paper's pretraining methods are relatively simple and can be applied to a broad class of problems. It is easy to imagine these methods being used by other members of the community.
- The paper's proposed ICL method is a creative emulation of attention mechanisms that is applicable to general neural operator methods.

**Weaknesses:**

I have concerns about the experimental design and worry the experimental results are overstated in the text. The claims about reducing data simulation costs are particularly overstated, and I believe the paper could be improved by contracting its stated goals to focus on real-world problem settings where data is scarce, and simulation methods are unavailable. I will list individual weaknesses below:
- The paper motivates the limited-data regime due to the computational burden of generating PDE solutions. The stated goal is to "reduce data simulation costs". However, the method suggests a pre-training procedure that trains full-sized models on $\geq 40,000$ samples for $500$ epochs. This also introduces a large computational cost, and this added cost of pre-training is never addressed. Without accounting for this added cost of pre-training, the paper's claims about reducing data simulation costs feel vacuous.
- The paper fails to contextualize their results on the PDE benchmarks with meaningful baselines. Figures 3 and 5 only consider random initialization of neural network weights and pretrained on a general computer vision task as baselines. This ignores the simple baseline of generating extra samples with the same computational cost as pretraining.
- Similarly, the paper fails to contextualize the results of the OOD experiment with meaningful baselines. The description of the OOD experiment indicates that the methods in citations [76, 77] are designed to solve the same problem. When reading the text, it seems like these methods could be used as baselines for comparison in Figure 6. If this is not the case, because [76, 77] are solving a weaker version of the problem or require significantly more data or compute resources, this is an advantage of the proposed method and should be highlighted in the paper.
- The amount and type of unlabeled pretraining data in the ERA5 and ScalarFlow experiments are unusual for a self-supervised learning setting, which raises questions about the significance of this experiment. I have written specific questions about this below.
- The results of the OOD experiment indicate the OOD task is still a difficult problem for neural operators, but the text does not discuss this difficulty. Without such a discussion, the significance of this experiment and the ICL method is unclear.

**Questions:**

General:
 - Line 132 states "initial snapshot $u_0(x)$ ... is sufficient for defining PDE systems." Could the authors explain or qualify this statement? The definition of a PDE system must also include a partial differential equation with parameters, a domain, and possibly boundary conditions like $u_0(x)$.
 - Line 185 states "Numerical solutions of PDEs are expected to exhibit invariance to filtering blur or different resolutions of inputs." Could the authors explain or qualify this statement? There are many examples where qualitative features of PDE solutions depend on the discretization level. One example is the Helmholtz problem considered with high wavenumber and forcing term with high-frequency oscillations. As the resolution of the input forcing term and solution increase, higher-frequency oscillations in the solution can be resolved. For the Poisson equation, one can show analytically that applying a blur to the forcing term will change the solution.
 - The experiment in Appendix E shows encouraging results that the general-purpose self-supervised learning method MoCo v2 does not perform well on the ERA5 task. I believe that if one were to combine Figure 8 and Figure 5a, the result would show that pretraining with MoCo v2 produces almost no improvement over random initialization. This seems like a nice result and wonder if the authors have intuition about why this is the case.

Questions about PDE experiments:
 - What are the inputs and outputs of the model for each experiment? As I was reading this paper, I was constantly asking this question and spent time jumping around in the appendix looking for details. Would it be possible to collect all of this information in a table for ease of reference?
 - For each PDE experiment, what is the distribution of forcing functions (Poisson, Helmholtz) or initial conditions (Reaction-Diffusion, Navier-Stokes)?
 - How does the "diffusion" physics parameter in Table 1 relate to the definition of the Poisson equation in Equation 1? Are elements of K randomly drawn from this range?
 - How does the Reynolds number relate to the definition of the Navier-Stokes equations (Equation 5)?
 - Are there spatial boundary conditions for the Reaction-Diffusion equation?
 - What do the error bars in Figures 3 and 4 represent? Are they showing the min/max of the three independent random seeds?

Questions about real-world data experiments:
 - The ERA5 and ScalarFlow datasets have temporal observations; the learning task is to predict the state of the system at time $t+16$ given a sequence of state observations at times $[t, t+15]$. What is the unlabeled data used in pretraining for these cases? Is it sequences of state observations at times $[t, t+15]$, for a set of different values $t$?
 - For these experiments, how does using snapshots at $t > 0$ agree with the definition of unlabeled PDE data for time-dependent systems in Section 3.1.1?
 - These experiments use a large number of snapshots to construct the pretraining dataset. In the ERA5 example, this is enough data to build $14,600 \times \frac{75}{100} \times \frac{1}{16} \approx 680$ labeled samples. In the ScalarFlow example, this is enough data to build $70,200 \times \frac{80}{100} \times \frac{1}{16} \approx 3500$ labeled samples. Could the snapshots used in the pretraining dataset be used in a supervised learning framework? If so, why doesn't the “random init” baseline have access to this data?
 - Are there examples of simulation or experimental settings where it is reasonable to assume we can easily gather snapshots of a system state, but these snapshots could not be used in a supervised neural operator framework? Mentioning such examples in the text would help readers understand the significance of this experiment.

Questions about the ICL method and OOD experiments:
 - Do the authors have any intuition for why the Poisson and Helmholtz OOD performance is poor, and the Navier-Stokes performance is better? Providing information about the types of PDE problems amenable to in-context learning would be a valuable contribution and would increase the impact of this experiment.
 - Is there a connection between the expected smoothness of a particular PDE’s solution and the performance of the ICL method on that PDE? For example, we know that the solution of the Poisson equation should be relatively smooth, but the proposed ICL method outputs highly non-smooth estimates.
 - I see that ICL with more demos improves the scale of the prediction. Are there problem settings where the scale of the solution is of particular interest? Discussing such settings may better contextualize the results on these challenging OOD tasks.


As they continue their work, I want to ensure the authors are aware of this preprint, which provides a fast method for generating synthetic data without a PDE solver, applicable to certain PDE problems. Fitting the neural operator to synthetic data could be a useful pretraining task if the PDE is known. I am NOT asking the authors to compare their method with this preprint:

E. Hasani and R. A. Ward, “Generating synthetic data for neural operators.” arXiv, Jan. 04, 2024. Available: http://arxiv.org/abs/2401.02398

**Limitations:**

The authors have addressed some limitations of their work. As mentioned above, I believe the computational expense of the pretraining stage is a limitation of this work which is not discussed in the paper.

---

> ### Author Rebuttal · Authors · 2024-08-07
>
> We truly thank the time and effort of reviewer u3i6 in reviewing our paper!
>
> > **Q1**. Without accounting for this added cost of pre-training, the paper's claims about reducing data simulation costs feel vacuous.
>
> Thanks for this great question!
> 1. We further provide extra experiments on joint unsupervised pretraining on multiple PDEs in **Figure 1 of our attached PDF response**. We can see that joint pretraining can further improve the performance of fine-tuning on different PDEs. This means fine-tuning on different PDEs can share the same pretrained checkpoint, which reduces the total computational costs. We will include this figure in our camera-ready version.
> 2. Recent works, like [1], also only considered per-PDE pretraining, even though their work is claimed to target SciML foundation models. Moreover, previous unsupervised pretraining works in computer vision also focused on a single dataset (like ImageNet), and they did not compare the costs of pretraining versus collecting more downstream samples.
> 3. The most important contribution of our paper is to define and leverage **unsupervised data** in PDEs and design **unsupervised pretraining** on top of it. Although pretraining is studied in scientific machine learning, **unsupervised pretraining is largely underexplored in the community**. Our work is the first kind, and will inspire the community in this research direction.
> In sum, we totally agree that reducing the total computation costs (training + sample collection) is important, and our promising joint pretraining results (**Figure 1 in our attached PDF response**) encourage us to keep working on this direction, but this should not weaken the core contribution and target in this paper.
>
>
> > **Q2**. Baseline of generating extra samples with the same computational cost as pretraining
>
> Actually, since we studied different numbers of samples in Figure 3, we can compare our method with baselines trained with extra generated samples (as indicated by the red arrows showing the savings in the number of samples).
>
>
> > **Q3**. Contextualize the results of the OOD experiment with meaningful baselines, such as [76, 77]
>
> [76, 77] by Liu et al. are related works, but their works cannot be directly compared with ours because: 1) they did not study 2D time-dependent PDEs; 2) they only scaled up to 5 demos, which is fewer than ours. We will highlight this more in our camera-ready version.
>
> Meanwhile, we further provide another baseline in the OOD in **Figure 3 in our attached PDF response**. In our paper, our ICL leverages the output (prediction) of neural operators to find similar samples (lines 251–255). In this new baseline, we try to use features extracted by the backbone of the neural operator (high-dimensional features before the final output layer) to find similar samples. As we can see, in general, this baseline is worse than our original ICL method, indicating that the final output of the neural operator can more accurately indicate true similar samples.
>
>
> > **Q4**. The results of the OOD experiment indicate the OOD task is still a difficult problem for neural operators, but the text does not discuss this difficulty.
>
> Thanks for the suggestion!
>
> Based on our visualization in Figure 14 in Appendix K, we believe OOD is challenging for neural operators because of:
> 1) Significantly different patterns of solutions under different physical parameters.
> 2) Different value ranges of solutions under different physical parameters.
> We will include this discussion in our camera ready.
>
>
> > **Q5**. Line 132 states "initial snapshot u0(x) ... is sufficient for defining PDE systems." Could the authors explain or qualify this statement?
>
> For a specific PDE with fixed physical parameters and a spatial-temporal domain, if we are given the initial condition and boundary condition, the dynamics of the PDE will be determined since there is no stochasticity in the PDE.
>
>
> > **Q6**. Line 185 states "Numerical solutions of PDEs are expected to exhibit invariance to filtering blur or different resolutions of inputs." Could the authors explain or qualify this statement?
>
> Our superresolution (SR) objective shares the same motivation as recent works on enhancing scientific data resolutions [2], which is to train SciML models to preserve the inherent physical properties of scientific data while learning the SR process. Given a ***specific*** input distribution, after memorizing the input data and fitting the SR objective, solutions of neural operators are expected to preserve the inherent physical properties and exhibit invariance to filtering blur. Moreover, in the context of pretraining, both objectives – MAE and SR – aim to further extract meaningful representations of input samples from current data distributions and provide a better-adapted initialization of network weights.
>
>
> > **Q7**. The experiment in Appendix E shows encouraging results that the general-purpose self-supervised learning method MoCo v2 does not perform well on the ERA5 task. ... This seems like a nice result and wonder if the authors have intuition about why this is the case.
>
> The main reason MoCo v2 does not perform well is that it was originally designed for object-centric images, and the loss is applied at the instance level (whole image) rather than at the pixel level (per spatial location), making it suboptimal for solving PDEs.

---

> ### Author Response · Authors · 2024-08-07
> **Continued Responses (1)**
>
> > **Q8**. What are the inputs and outputs of the model for each experiment? As I was reading this paper, I was constantly asking this question and spent time jumping around in the appendix looking for details. Would it be possible to collect all of this information in a table for ease of reference?
>
> Thanks for this suggestion! Here we include another table of details (input/output shapes). Due to the page limit, we will try to squeeze this table into our camera ready.
> |  | Input | Input Shape | Output |
> |:---:|:---:|:---:|:---:|
> | Poisson | source function (f), three diffusion coefficients | CxHxW (C = 4) | potential field (u) |
> | Helmholtz | source function (f), wavenumber | CxHxW (C = 2) | wave function (u) |
> | NS (FNO) | vorticity (w) | TxHxW (T=33) | vorticity (w) at T+1 |
> | NS (PDEBench) | velocity (Vx, Vy), vorticity (w) | TxCxHxW (T=15, C=3) | velocity (Vx, Vy), vorticity (w) at T+1 |
> | RD (PDEBench) | activator (u), inhibitor (v) | TxCxHxW (T=10, C=2) | activator (u), inhibitor (v) |
>
>
> > **Q9**. For each PDE experiment, what is the distribution of forcing functions (Poisson, Helmholtz) or initial conditions (Reaction-Diffusion, Navier-Stokes)?
>
> For generating forcing functions or initial conditions, we follow previous works [1, 3, 4] and adopt their numerical solvers. We further design the ranges of physical parameters as listed in Table 1.
>
>
> > **Q10**. How does the "diffusion" physics parameter in Table 1 relate to the definition of the Poisson equation in Equation 1? Are elements of K randomly drawn from this range?
>
> The "diffusion" physics parameter in Table 1 determines the eigenvalue of the diffusion coefficient tensor. For more details about the construction of diffusion coefficient tensors, please consider reading the paragraph “PDE coefficient sampling” in Section 3 of [1].
>
>
> > **Q11**. How does the Reynolds number relate to the definition of the Navier-Stokes equations (Equation 5)?
>
> The Reynolds number is defined as $Re=\frac{\rho u L}{\nu}$, where $\nu$ is the viscosity of the fluid ($\rho$ is the density of the fluid, $u$ is the flow speed, $L$ is a constant of the fluid’s characteristic linear dimension). $\nu$ is used in Eq. 5. In other words, a smaller $\nu$ indicates a larger Reynolds number.
>
>
> > **Q12**. Are there spatial boundary conditions for the Reaction-Diffusion equation?
>
> Following PDEBench [3], we consider the Neumann boundary condition for Reaction-Diffusion.
>
>
> > **Q13**. What do the error bars in Figures 3 and 4 represent?
>
> Error bars in Figures 3 and 4 indicate standard deviations of three independent runs with different random seeds.
>
>
> > **Q14**. The ERA5 and ScalarFlow datasets have temporal observations; the learning task is to predict the state of the system at time t+16 given a sequence of state observations at times [t,t+15]. What is the unlabeled data used in pretraining for these cases? Is it sequences of state observations at times [t,t+15], for a set of different values t?
>
> Similar to time-dependent PDEs (reaction-diffusion, Navier-Stokes), which are explained in line 131 and in Appendix A, on ERA5 and ScalarFlow, models are trained on individual snapshots without temporal information. The pretraining, fine-tuning, and test sets are separate without overlap. We will provide clearer explanations in our camera-ready version.
>
>
> > **Q15**. For these experiments, how does using snapshots at t>0 agree with the definition of unlabeled PDE data for time-dependent systems in Section 3.1.1?
>
> In time-dependent PDEs, the simulation can start from any snapshot (i.e., the numerical simulation can take any snapshot as the initial condition and continue simulating the dynamics). In fact, in FNO, the actual simulation of Navier-Stokes starts after the fluid roughly exits the chaotic phases (please refer to the GitHub repo called `neuraloperator/physics_informed`, line 34 of `generate_data.py`).
>
> This setting is also similar to starting the collection of climate data at any timestep and continuing the collection. The key is that we do not involve any temporal dynamics during our pretraining.
>
>
> > **Q16**. Could the snapshots used in the pretraining dataset be used in a supervised learning framework? If so, why doesn't the “random init” baseline have access to this data?
>
> The “random init” baseline can access this data. However, as we consider these pretraining samples as individual snapshots (to simulate the practice where people cannot capture temporal dynamics and can only collect individual spatial climate/flow snapshots), we do not use any temporal information in these samples, which cannot be used for supervised forecasting.

---

> ### Author Response · Authors · 2024-08-07
> **Continued Responses (2)**
>
> > **Q17**. Are there examples of simulation or experimental settings where it is reasonable to assume we can easily gather snapshots of a system state, but these snapshots could not be used in a supervised neural operator framework?
>
> Collecting snapshots with temporal dynamics in large-scale scenes is significantly more challenging than collecting individual snapshots without temporal dynamics. We have the following examples, and we will try to include them in our camera ready:
> 1. Weather forecasting: Continuous monitoring of atmospheric parameters requires a network of weather stations, satellites (e.g. large amount of historical satellite data was leveraged in ERA5), and real-time data processing, whereas a single weather report is simpler and less resource-intensive. Long-term climate monitoring involves maintaining observation networks for years, compared to a one-time collection of current climate conditions.
> 2. Smoke dispersion studies: Tracking smoke spread over time requires multiple sensors (e.g. five cameras with carefully calibrated intrinsics and extrinsics in ScalarFlow) and continuous data processing, unlike a single air quality measurement at a peak pollution event.
> 3. Ocean currents: The monitoring demands continuous deployment of buoys and sensors, whereas a single measurement of water conditions can be done with minimal equipment like a handheld sensor or a small research boat.
>
>
> > **Q18**. Do the authors have any intuition for why the Poisson and Helmholtz OOD performance is poor, and the Navier-Stokes performance is better? Is there a connection between the expected smoothness of a particular PDE’s solution and the performance of the ICL method on that PDE? For example, we know that the solution of the Poisson equation should be relatively smooth, but the proposed ICL method outputs highly non-smooth estimates.
>
> To understand the results in the OOD setting, we provide a visualization in Figure 14 in Appendix K. We find that the Helmholtz equation in the OOD setting exhibits much more complicated unseen patterns than the Poisson and Navier-Stokes equations, making it challenging to learn.
>
> The non-smoothness of the ICL method outputs is mainly due to the uncertainty of neural operators on out-of-distribution (OOD) samples. As the sample-wise similarity is measured by the neural operators’ outputs, which are not accurate, the more severe the OOD setting, the less confident the model’s output and similarity measurement will be.
>
>
> > **Q19**. I see that ICL with more demos improves the scale of the prediction. Are there problem settings where the scale of the solution is of particular interest? Discussing such settings may better contextualize the results on these challenging OOD tasks.
>
> Thanks for this suggestion! In numerical simulations or predictions of PDEs, there are settings where the scale or magnitude of solutions is more important than the exact shape/pattern. We will try to include these examples in our camera ready.
> 1. Heat Transfer: In large-scale systems, the focus might be on overall temperature and extreme values. For instance, in evaluating a cooling system, the key concern might be the peak temperature rather than the detailed temperature distribution.
> 2. Fluid Dynamics: For applications like aerodynamics, the overall drag or lift force on an object is often more critical than capturing every detail of the flow pattern, such as in airfoil design.
> 3. Environmental Modeling: The concentration of pollutants at specific locations or total pollutant transport is often more crucial than the exact distribution, such as in groundwater flow studies.
>
>
> > **Q20**. Regarding related work: E. Hasani and R. A. Ward, “Generating synthetic data for neural operators.” arXiv, Jan. 04, 2024.
>
> Thanks for bringing this up! We noticed this work after our paper submission. Our method, which works on unlabeled PDE data, is orthogonal to their generation of synthetic PDE data, and the two approaches could potentially be combined into a semi-supervised learning strategy.
>
> [1] “Towards Foundation Models for Scientific Machine Learning: Characterizing Scaling and Transfer Behavior“ Subramanian et al. 2023
>
> [2] “SuperBench: A Super-Resolution Benchmark Dataset for Scientific Machine Learning” Ren et al. 2023.
>
> [3] “PDEBENCH: An Extensive Benchmark for Scientific Machine Learning“ Takamoto et al. 2022
>
> [4] “Fourier Neural Operator for Parametric Partial Differential Equations” Li et al. 2020

---

> ### Comment · Reviewer_u3i6 · 2024-08-09
>
> Many thanks to the authors for providing detailed responses to all parts of the review. I have specific comments and questions about a few parts of the response.
>
> **Q1 and Q9** I really appreciate the design of this extra experiment, and the results look encouraging. However, the lack of detail about the distribution of forcing functions and source functions still obfuscates the significance of these results. I have briefly read [1] to understand the distribution of forcing functions and source functions. To my understanding, the distribution of Helmholtz and Poisson source functions are the same. Is this correct?
>
> **Q2** In my opinion, Figure 3 and the relevant text do not give enough detail to compare the proposed method with the baseline of generating extra samples. This is because there is not enough information to evaluate how long it takes to pretrain the model (I could not find this in Section B3), and there is no information about how long it takes to generate samples for the Helmholtz or Poisson problems.
>
> **Q4** Thanks for this explanation. Do you plan to update any of the relevant text in the camera-ready version?
>
> **Q6** Thanks for this explanation. The explanation makes sense but is quite different from what appears in the original submission. Do you plan to update this in the camera-ready version?
>
> **Q8** Could you also add the shapes of the pretraining inputs/outputs to this table?
>
> Thank you for all of the other replies; again, I really appreciate the careful and thorough response.

---

> > ### Author Response · Authors · 2024-08-09
> > **Thank you very much for your reply!**
> >
> > We truly thank reviewer u3i6 for reading our response and providing the prompt reply!
> >
> > Below are our responses to your further questions:
> >
> > > **Q1**. Distribution of Helmholtz and Poisson source functions.
> >
> > Yes, the source distributions are the same for Poisson and Helmholtz. However, they have different ranges of physical parameters in their input, and they have different numbers of input channels (4 for Poisson, 2 for Poisson).
> >
> >
> > > **Q2**. Pretraining costs and simulation costs.
> > 1. Pretraining costs. FNO: 20 GPU hours. Video-MAE: 18 GPU hours. (GPU: A100)
> > 2. Simulation costs (8192 downstream samples for Figure 3):
> > * Poisson: 0.04 hour
> > * Helmholtz: 7.85 hour
> >
> > > **Q3**. Update new explanations into camera ready.
> >
> > Yes, we promise to update the explanations for both your previous Q4 and Q6. We thank you for this meaningful discussion!
> >
> >
> > > **Q4**. Shapes of the pretraining inputs/outputs.
> >
> > Thanks for the suggestion! We put the shape of our input/output during pretraining into the table below. Since our pretraining is based on reconstruction, the input and output share the same shape.
> > | PDE | Input/Output (Reconstruction) | Shape |
> > |---|:---:|:---:|
> > | Poisson | source function (f), three diffusion coefficients | CxHxW (C = 4) |
> > | Helmholtz | source function (f), wavenumber | CxHxW (C = 2) |
> > | NS (FNO) | vorticity (w) | HxW |
> > | NS (PDEBench) | velocity (Vx, Vy), vorticity (w) | CxHxW (C=3) |
> > | RD (PDEBench) | activator (u), inhibitor (v) | CxHxW (C=2) |

---

> > > ### Comment · Reviewer_u3i6 · 2024-08-12
> > >
> > > Thank you very much for the extra information.
> > >
> > > I have a few more questions about the joint pretraining experiment:
> > >  - How are the mismatched input dimensions handled? As you point out, the Helmholtz and Poisson data have different numbers of channels. How do you train one network to handle both types of data?
> > >  - When you do the joint pretraining, does the model see all 46,080 unlabeled Poisson samples and all 46,080 unlabeled Helmholtz samples?
> > >  - The caption of Figure 2 in the rebuttal document says Poisson and Helmholtz problems have different initial conditions. Should this say different distributions of physics parameters instead?
> > >
> > > The information about the pretraining and simulation costs is encouraging. After seeing these numbers, it is clear that pretraining with your method is a more effective use of compute resources than generating new samples. I believe the paper would be strengthened if this information is included. This was my major concern; I am happy to raise my score as it has been addressed.
> > >
> > > As a minor note, I think your estimate for the simulation costs for the Helmholtz equation seems quite high. I believe discretizing the Helmholtz operator + periodic boundary conditions gives a very sparse linear operator which can be inverted quite quickly using standard sparse linear algebra software.

---

> > > > ### Author Response · Authors · 2024-08-12
> > > > **Thank you very much for your reply!**
> > > >
> > > > We truly thank reviewer u3i6 for reading our response and providing this further reply!
> > > >
> > > > > **Q1**. How are the mismatched input dimensions handled?
> > > >
> > > > For Helmholtz inputs, we pad extra channels with zeros to match inputs of Poisson.
> > > >
> > > >
> > > > > **Q2**. When you do the joint pretraining, does the model see all 46,080 unlabeled Poisson samples and all 46,080 unlabeled Helmholtz samples?
> > > >
> > > > * The number of pretraining samples: We used all 46,080 unlabeled Poisson samples and all 46,080 unlabeled Helmholtz samples. We decided to choose this setting because related works on SciML foundation models [1,2] also used all samples from each PDE for pretraining.
> > > > * Pretraining costs: We halved the training steps (epochs) to make the total pretraining cost equivalent to pretraining on single PDEs and make the comparison fair.
> > > >
> > > >
> > > > > **Q3**. The caption of Figure 2.
> > > >
> > > > Thanks for this suggestion! Yes, we will make this caption more precise by noting that they have different distributions of physics parameters in our camera-ready version.
> > > >
> > > >
> > > > > **Q4**. Simulation costs for the Helmholtz.
> > > >
> > > > Thanks for this suggestion! We are definitely open to further accelerating our simulators. For generating Helmholtz samples, we follow `utils/gen_data_helmholtz.py` in the codebase of [3].
> > > >
> > > > [1] “Multiple Physics Pretraining for Physical Surrogate Models” McCabe et al. 2023
> > > >
> > > > [2] “DPOT: Auto-Regressive Denoising Operator Transformer for Large-Scale PDE Pre-Training” Hao et al. 2024.
> > > >
> > > > [3] “Towards Foundation Models for Scientific Machine Learning: Characterizing Scaling and Transfer Behavior“ Subramanian et al. 2023

---

### Official Review · Reviewer_6vSy · 2024-07-11

**Soundness:** 3
**Presentation:** 3
**Contribution:** 2
**Rating:** 6
**Confidence:** 3

**Summary:**

This paper proposes a pre training strategy for operator learning. They introduce unsupervised training in the case of physical data. 2 architectures are studied: transformers and FNO. Finally, some experiments on different PDEs are conducted to highlight the properties of the model.

**Strengths:**

The paper is well-written and explains step-by-step the algorithms and architecture.
Several experiments highlight the performances of the model on both simulated and real-world datasets, which is a challenging setting. Moreover, statistics are provided for each experiment.

**Weaknesses:**

As mentioned in the paper, masked auto-encoding has already been proposed in CV applications. The contribution however is unclear for me. ICL and masked-auto-encoding are widely known techniques. Moreover, it is difficult to identify why is the unsupervised pretraining performs best than other baselines.

**Questions:**

-	For the unlabeled data for time-dependent PDE, how does the PDE ICs are sufficient to identify the PDE? How does the model have access to the PDE parameters?
-	Is your model trained on multiple physics? Or pretraining are proceeded on one PDE?
-	On figure3, why are the unsupervised training performing better than training from scratch? How is proceeded this “random init” training? Is it using data with supervised loss?
-	What is the comparison of other baselines in the OOD setting?
-	What is the computational cost of training your model compared to other baselines? How many parameters do the baselines contains?

**Limitations:**

Author have proposed a discussion on the limitations of their paper. However, they did not address societal impact and ecological considerations.

---

> ### Author Rebuttal · Authors · 2024-08-07
>
> We truly thank the time and effort of reviewer 6vSy in reviewing our paper!
>
> > **Q1**. Masked auto-encoding has already been proposed in CV applications. The contribution however is unclear for me.
>
> We would like to clarify and re-emphasize: The most important contribution of our paper is to define and leverage **unsupervised data** in PDEs and design **unsupervised pretraining** on top of it. This is mentioned in our first contribution bullet (lines 66–69) and also in line 7 of our abstract. There are previous works that try to pretrain neural operators on simulated solutions [1, 2, 3], but no previous works tried to reduce the simulation costs by defining and leveraging unlabeled PDE data. That means, although pretraining is studied in scientific machine learning, **unsupervised pretraining is largely underexplored in the community**. Our work is the first kind, and will inspire the community in this research direction.
>
> We do not claim that our MAE method is new, and we also provide clear motivation for why we introduce MAE in pretraining neural operators (lines 160–172).
>
>
> > **Q2**. Moreover, it is difficult to identify why is the unsupervised pretraining performs best than other baselines.
>
> > (Similar question) On figure3, why are the unsupervised training performing better than training from scratch?
>
> We discussed the benefits of our unsupervised pretraining in Figure 4 in Sec. 4.1. There are three reasons:
>
> 1. Reduced overfitting, which consistently leads to smaller generalization gaps across all PDEs we studied.
> 2. Faster convergence, which can accelerate model convergence more than both random initialization and vision-pretrained checkpoints.
> 3. Meaningful representations, which are extracted from pretrained Video-MAE and are helpful during fine-tuning.
>
>
> > **Q3**. For the unlabeled data for time-dependent PDE, how does the PDE ICs are sufficient to identify the PDE? How does the model have access to the PDE parameters?
>
> Our models do not have access to physical parameters. Given the initial condition and boundary condition, the dynamics of a PDE are determined since there is no stochasticity in the PDE.
>
>
> > **Q4**. Is your model trained on multiple physics? Or pretraining are proceeded on one PDE?
>
> Our experiments focus on pretraining on one single PDE.
>
> However, we also further studied joint pretraining. We provide extra experiments in **Figure 1 in our attached PDF response**. We can see that joint pretraining can further improve the performance of fine-tuning on different PDEs. We will include this figure in our camera-ready version.
>
>
> > **Q5**. How is proceeded this “random init” training? Is it using data with supervised loss?
>
> In Figure 3, the “random init” and “unsupervised” models share completely the same fine-tuning settings, and they both use the supervised loss (labeled PDE data) during fine-tuning.
>
> The only difference is that “random init” uses random initialization of model weights to start the fine-tuning, while “unsupervised” uses pretrained weights (from our unsupervised pretraining) to start the fine-tuning.
>
>
> > **Q6**. What is the comparison of other baselines in the OOD setting?
>
> Thanks for the suggestion!
>
> We further provide a baseline in the OOD setting, attached in **Figure 3 in our attached PDF response**.
>
> In our paper, our ICL method leverages the output (prediction) of neural operators to find similar samples (lines 251–255). In this new baseline, we try to use features extracted by the backbone of the neural operator (high-dimensional features before the final output layer) to find similar samples. As we can see, in general, this baseline is worse than our original ICL method, indicating that the final output of the neural operator can more accurately indicate true similar samples.
>
>
> > **Q7**. What is the computational cost of training your model compared to other baselines? How many parameters do the baselines contains?
>
> * Pretraining costs. FNO: 20 GPU hours. Video-MAE: 18 GPU hours. (GPU: A100)
> * Model parameters. FNO: 67.1M. Video-MAE: 23.4M.
>
>
> > **Q8**. Societal impact and ecological considerations
>
> Our paper improves the data efficiency of neural operators for solving PDEs, which can significantly reduce computational costs and energy consumption associated with high-fidelity numerical PDE simulations. This improvement can lead to more sustainable scientific research by minimizing the environmental footprint of extensive computational experiments. Moreover, by democratizing access to advanced PDE solutions through more efficient pretraining, our method has the potential to accelerate scientific and engineering advancements in the broader community, benefiting society at large.
>
> [1] “Lie Point Symmetry Data Augmentation for Neural PDE Solvers” Brandstetter et al. 2022
>
> [2] “Self-supervised learning with lie symmetries for partial differential equations“ Mialon et al. 2023
>
> [3] “Towards Foundation Models for Scientific Machine Learning: Characterizing Scaling and Transfer Behavior“ Subramanian et al. 2023

---

> > ### Comment · Reviewer_6vSy · 2024-08-11
> >
> > I thank the reviewer for their complete answers to my questions. Based on their answers, I still have other questions/clarifications.
> >
> > - **Q2**: How do you prove these 3 points (experimental). I think fig. 3 and the OOD experiment answer point n1. But how faster is your method compared to others? Is it in term of computational time or number of steps?
> > - **Q3**: I am still not sure to fully understand this point. If your model doesn't have access to the physics and the parameters, how does the model learn the dynamics? Given fixed ICs and/or BCs, several differential operator could be applied to evolve the trajectory. Moreover, you state that you model doesn't have access to the physical parameters. However, on the rebuttal pdf, the legend states that input dimension are different for the reasons of different physical parameters.
> > - **Q4**: thanks for the additional experiment, which I believe is the core question for pre training.
> > - **Q5** thanks for the clarification. This behavior is to be expected since the *pretrained model* have seen more examples than the *random init* one? Are the examples during pertaining the same as the one used during fine-tuning?
> > - **Q8**: I agree with your statement, however, pre training on a single PDE is, I think, very costly, and I would be curious of how long would it take to have a real impact?

---

> > > ### Author Response · Authors · 2024-08-11
> > > **Thank you very much for your reply!**
> > >
> > > We truly thank reviewer 6vSy for reading our response and providing the reply!
> > >
> > > Below are our responses to your further questions:
> > >
> > > > **Q2**. How do you prove these 3 points (experimental).
> > >
> > > We would appreciate it if reviewer 6vSy could kindly read our Figure 4 (main submission) and line 292-303. Three subplots in Figure 4 correspond to our three points.
> > >
> > >
> > > > **Q3**. If your model doesn't have access to the physics and the parameters, how does the model learn the dynamics? Moreover, you state that you model doesn't have access to the physical parameters.
> > >
> > > 1. In data-driven operator learning, neural operators do not necessarily need to access physical parameters (see [1, 2]). This is mainly because the training PDE data is mostly sampled from a fixed distribution of physical parameters, and the neural operator is trained as a numerical solver for this fixed distribution. This is also why neural operators could perform poorly on out-of-distribution samples.
> > > 2. To be more precise, on Reaction-Diffusion and Navier Stokes, FNO and VMAE do not have access to physical parameters. To make this more clear, here we include a table of detailed input/output shapes. Due to the page limit, we will try to squeeze this table into our camera ready.
> > > |  | Input | Input Shape | Output |
> > > |:---:|:---:|:---:|:---:|
> > > | Poisson | source function (f), three diffusion coefficients | CxHxW (C = 4) | potential field (u) |
> > > | Helmholtz | source function (f), wavenumber | CxHxW (C = 2) | wave function (u) |
> > > | NS (FNO) | vorticity (w) | TxHxW (T=33) | vorticity (w) at T+1 |
> > > | NS (PDEBench) | velocity (Vx, Vy), vorticity (w) | TxCxHxW (T=15, C=3) | velocity (Vx, Vy), vorticity (w) at T+1 |
> > > | RD (PDEBench) | activator (u), inhibitor (v) | TxCxHxW (T=10, C=2) | activator (u), inhibitor (v) |
> > >
> > >
> > > > **Q5**. This behavior is to be expected since the pretrained model have seen more examples than the random init one? Are the examples during pertaining the same as the one used during fine-tuning?
> > >
> > > 1. This behavior (“random init” is worse than “unsupervised pretrained”) is mainly because of three benefits we discussed in Figure 4: reduced overfitting, faster convergence, and meaningful representations.
> > >
> > > 2. Examples during pretaining are different from one used during fine-tuning: no overlap.
> > >
> > >
> > > > **Q8**. Pre training on a single PDE is, I think, very costly, and I would be curious of how long would it take to have a real impact?
> > >
> > > Thanks for this question!
> > >
> > > Pretraining costs. FNO: 20 GPU hours. Video-MAE: 18 GPU hours. (GPU: A100)
> > >
> > > We are working with domain scientists to achieve real impacts on scientific problems as soon as possible (representative examples are climate/weather/airfoil in our Figure 5, where our method has clear improvements).
> > >
> > > Meanwhile, we would like to emphasize:
> > > 1. Our extra experiments in Figure 1 of our PDF response indicate that **joint unsupervised pretraining on multiple PDEs can reduce the total computational costs**, because fine-tuning on different PDEs can share the same pretrained checkpoint.
> > >
> > > 2. **Recent works, like [3], also only considered per-PDE pretraining**, even though their work is claimed to target SciML foundation models. Moreover, previous unsupervised pretraining works in computer vision also focused on a single dataset (like ImageNet), and they did not compare the costs of pretraining versus collecting more downstream samples.
> > >
> > > We hope the above responses have addressed your concerns. We are happy to discuss this further!
> > >
> > >
> > > [1] “Multiple Physics Pretraining for Physical Surrogate Models” McCabe et al. 2023
> > >
> > > [2] “PDEBENCH: An Extensive Benchmark for Scientific Machine Learning“ Takamoto et al. 2022
> > >
> > > [3] “Towards Foundation Models for Scientific Machine Learning: Characterizing Scaling and Transfer Behavior“ Subramanian et al. 2023

---

> > > > ### Comment · Reviewer_6vSy · 2024-08-12
> > > >
> > > > I thank authors for their complete and fast answer to my (numerous) questions.
> > > >
> > > > -**Q3**: 1. Indeed NO and data-driven method do not have access to physical parameters, however, they have access to the dynamics of the PDE through samples seen during training. From what I understood in your paper, this is not the case of your model, which is unsupervised. This explains my confusion on this point.
> > > > 2. Thanks a lot for this table that helps me understand the input.outputs. For static PDE, PDE coefficient and BC could be sufficient to identify the PDE. However, I am confused about time-dependent PDEs: if only ICs are given as input, and that no simulation were proceed, how does the model learns to forecast the next tilmestep?

---

> > > > > ### Author Response · Authors · 2024-08-12
> > > > > **Thank you very much for your reply!**
> > > > >
> > > > > We truly thank reviewer 6vSy for reading our response and providing this further reply!
> > > > >
> > > > > For time-dependent PDEs:
> > > > > * **Pretraining** Phase: neural operators only access individual snapshots, and you are correct: during pretraining, our neural operators only learn spatial reconstruction without learning any temporal dynamics.
> > > > > * **Fine-tuning** Phase: our neural operators access snapshots with temporal dynamics, and learn both spatial information and forecast the next temporal steps.
> > > > >
> > > > > We hope this response resolves your further question. We are happy to discuss further!

---

> > > > > > ### Comment · Reviewer_6vSy · 2024-08-13
> > > > > >
> > > > > > I thank again the author for their answer and clarifications. This means that for time dependent PDE, data are needed during the fine-tuning stage in order to learn the temporal dynamics.
> > > > > > I will raise my score to 6 since all my questions have been answered.

---

### Official Review · Reviewer_oEEz · 2024-07-11

**Soundness:** 3
**Presentation:** 2
**Contribution:** 2
**Rating:** 5
**Confidence:** 4

**Summary:**

This paper aims at improving the data efficiency of deep learning models for tackling Operator Learning. The paper focuses on two aspects: 1) They pretrain neural operators on data that do not assume labels, i.e. without the target function or the trajectory solution of states. To do so, they rely on a masked auto-encoding task or a super-resolution task, that both can be trained without labels. They show that this pretraining strategy can effectively improve the sample efficiency compared to a random initialization.  2) They propose an algorithm to do in-context learning at test time based on a number of demo examples and show results on out-of-domain PDEs.

**Strengths:**

* The main strength of the paper is that the pretraining strategy effectively reduces the number of samples to reach a certain target accuracy compared to a random initialization.
* The method is tested against multiple equations from different levels of difficulty.
* The comparison with pretrained vision transformers, though a little counterintuitive, is informative on the capacity of transferability of this kind of models to different tasks.
* The ICL experiments seem to be consistent, the more demos the better the results.

**Weaknesses:**

* Overall the paper is quite difficult to read, there are many different aspects that are described within the method section and it can be hard to understand in the experiment section which is what.
* If I am not mistaken there is no consistent comparison between the different tasks and architectures of the method (super-resolution vs masking x FNO vs transformer) for the different datasets.
* The ICL method should be compared to at least one baseline. I suggest to do a simple kernel method but on the inputs rather than with the predictions of FNO. It can be a kNN or anything, but it would actually give a good understanding of how well the method works.

**Questions:**

* What is the best effective strategy for pretaining ? Is it the masking-damasking or super-resolution that yields the best results ? Or do you use both ?
* Did you observe a difference between the video transformer and FNO for this setup ? Is there one that you would recommend better than the other ?

**Limitations:**

There is a limitation section.

---

> ### Author Rebuttal · Authors · 2024-08-07
>
> We truly thank the time and effort of reviewer oEEz in reviewing our paper!
>
> > **Q1**. Overall the paper is quite difficult to read
>
> Thanks for this suggestion! We will try to make our teaser figure (Figure 1) clearer and connect it more to the subsections in methods and experiments, so readers can follow easily.
>
>
> > **Q2**. Comparison between the different tasks and architectures of the method
>
> 1. To focus on pretraining (model initialization), we keep the network architecture the same in each experiment to ensure that our comparisons in each task are fair.
> 2. We study the contribution of masking vs. super-resolution in Appendix G.1. We find that when training with a low volume of data, we should use much stronger perturbations (high masking ratios and strong blur), whereas a high volume of data only requires mild perturbations.
>
>
> > **Q3**. The ICL method should be compared to at least one baseline.
>
> Thanks for the suggestion!
>
> We further provide a baseline for the ICL method in **Figure 3 in our attached PDF response**.
>
> In our paper, our ICL leverages the output (prediction) of neural operators to find similar samples (lines 251–255). In this new baseline, we try to use features extracted by the backbone of the neural operator (high-dimensional features before the final output layer) to find similar samples. As we can see, in general, this baseline is worse than our original ICL method, indicating that the final output of the neural operator can more accurately indicate true similar samples.
>
>
> > **Q4**. What is the best effective strategy for pretaining? Is it the masking-damasking or super-resolution that yields the best results? Or do you use both?
>
> We find that combined methods (MAE + superresolution) give the best performance. These results are in Appendix G.1, where we exhaustively studied the choices of hyperparameters for MAE and superresolution.
>
> > **Q5**. Did you observe a difference between the video transformer and FNO for this setup?
>
> We tend to avoid recommending specific model architectures, as our method is agnostic to the architecture choices. We have proven that our method can help both FNO and transformers.

---

> > ### Comment · Reviewer_oEEz · 2024-08-09
> > **Response**
> >
> > Thank you for your response.
> >
> > I believe the pretraining strategy is interesting, especially with the notion of unlabelled data. On the other hand, I am not convinced by the results of the "in-context" learning experiments. While the loss effectively decreases with the number of demo samples, and while using FNO seems to produce better results than with the backbone features (on navier-stokes only), it appears that the gap on Poisson and Helmholtz datasets between in-context and classical experiments is significant (relative errors close to 1 for ICL).  As a result I do not think that the ICL method presented here is effectively taking advantage from the demos.
> >
> > I will maintain my score and suggest the authors to only focus their contribution on the pretraining strategy.

---

> > > ### Author Response · Authors · 2024-08-09
> > > **Thank you very much for your reply!**
> > >
> > > We truly thank reviewer oEEz for reading our response and providing the prompt reply!
> > >
> > > We understand your concern about our ICL experiments. Meanwhile, we would like to emphasize the following:
> > >
> > > 1. The **purpose of ICL** is to **continuously reduce the test error with more demos**. This is demonstrated in our figures, where the test error curve drops for Poisson, Helmholtz, and Navier-Stokes. Additionally, we have shown that our method can **scale up to a much larger number of demos** compared to [1].
> > >   - Furthermore, we demonstrate that our ICL method can further reduce the model's uncertainty (evidenced by shorter error bars) as more demos are used. This reduction in uncertainty was not shown in [1].
> > >
> > > 2. The **value of error** is mainly determined by the **domain gap** between the training source data and the unseen testing data.
> > > Since we chose to test our ICL in an out-of-distribution (OOD) setting, this domain gap is very large (as indicated by the error when #demos = 0).
> > >   - In fact, **our OOD performance is aligned with previous works**, and it is well-known that the performance of neural operators in OOD settings is poor and challenging. For reference, see **subplot (f) in "Figure B.2: Addressing (Q4)" of [2]** (in Appendix B.2, at the very bottom of the paper): Performance on Helmholtz (“SYS-3”) is poor (with errors close to 1 for both their model and ours), even when they fine-tuned with supervision on a few OOD samples.
> > >
> > > In sum, we are open to further improving the performance of our ICL method. Meanwhile, we have demonstrated that our ICL method advances beyond previous works [1].
> > >
> > > [1] "In-context operator learning with data prompts for differential equation problems" Liu et al. 2023
> > >
> > > [2] “Towards Foundation Models for Scientific Machine Learning: Characterizing Scaling and Transfer Behavior“ Subramanian et al. 2023

---

### Official Review · Reviewer_oUb1 · 2024-07-13

**Soundness:** 2
**Presentation:** 2
**Contribution:** 2
**Rating:** 5
**Confidence:** 4

**Summary:**

This paper presents an unsupervised pretraining approach for PDE solvers based on Meta Autoencoding and Super Resolution. The authors show that after pretraining, the model can achieve better accuracy than training a solver from scratch. This paper also presents an "in-context learning" strategy for inference on out-of-distribution data.

**Strengths:**

- Designing pretraining techniques for SciML foundation model is an important research direction.

- To the best of my knowledge, the proposed pretraining technique is new in the SciML context.

**Weaknesses:**

- **Some important experimental setups are missing**. In the main body of the paper, it's not even clear what the model size is, what the pretraining datasets are, and whether the model is pretrained on one class of PDEs or multiple classes of PDEs.
  - A follow-up concern is that according to Table 2 in Appendix B, it seems the authors pretrain different models for different PDEs. This significantly undermines the value of pretraining, because the ultimate goal of pretraining is to obtain a general purpose foundation model which can be easily transferred to different tasks (i.e., different PDEs in this context). If the method requires pretraining for each single class of PDEs, it's not efficient any more, and the comparison in Fig. 3 is not fair because **the proposed approach is still a problem-specific technique** and the pretrained models require additional training.

- **Missing baselines.** For time-independent PDEs, the MAE for image should also be included as a baseline to support the claim of _outperforming conventional vision-pretrained models_.

- I feel that **"in-context learning" in Sec. 3.2 is a misnomer**. First, there is no notion of "context" in the pretrained model. Second, the presented model is not an auto-regressive model. Third, unlike the self-attentive models, the presented model does not actively or flexibly "learn" from the examples; the way to aggregate information is fixed given Algorithm 1. The presented "ICL" algorithm deviates too much from the standard practice. I strongly encourage the authors to get rid of  "ICL" and refers the proposed approach as a few-shot learning strategy in the paper to avoid misleading the community.

- In Sec. 4.3, the few-shot learning accuracy looks poor, and the standard deviation of the error is not reported.

- Some discussions in the papers are inaccurate or lack justification.
  - Line 95: BERT does not leverage next token prediction for pretraining.
  - Line 142: Multiple benefits of pretraining are listed, but no references are provided, and no relevant experiments are presented to justify those arguments (e.g., better generalization, faster training speed, etc).
  - Line 158: the authors cite the BERT paper and claim that "_these methods enable training in NLP and CV of generalizable models containing over one hundred billion parameters_". I don't think any variant of BERT is scaled to one hundred billion parameters.

- Other issues.
  - Line 26: DNN $\to$ deep neural network
  - Line 32: has  $\to$  have
  - Line 46: start   $\to$  start with

**Questions:**

- Can you report the model sizes, pretraining cost, and finetuning cost?

- Can the pretrained model be applied to other PDEs which are not seen in pretraining?

**Limitations:**

The authors discuss some of the limitations of this paper in Sec. 6.

---

> ### Author Rebuttal · Authors · 2024-08-07
>
> We truly thank the time and effort of reviewer oUb1 in reviewing our paper!
>
> Before providing detailed responses, we would like to make a **clarification**:
>
> Since our paper focuses on **unsupervised pretraining** *but not* SciML foundation models (we *never* claimed our pretraining leads to such models), we hope reviewer oUb1 will not assume we are pursuing or comparing with the latest SciML foundation model papers. We believe it is *unfair* to ask us to match their experimental settings.
>
> As the benefits and strategies of pretraining for SciML are still under-explored, both supervised/unsupervised/per-PDE/joint pretraining are meaningful to this research direction. This is similar to what researchers in the computer vision community explored with single dataset/task pretraining [1, 2, 3] before moving on to foundation models.
>
> ---
> ---
>
> > **Q1**. Some important experimental setups are missing
>
> 1. Model sizes. FNO: 67.1M. Video-MAE: 23.4M.
> In each experiment, all compared models are of the same architecture and thus fairly compared.
> 2. Details about pretraining datasets are clearly documented in Appendix A, which is also clearly referred to in Line 134.
> 3. In each experiment, we pretrain our model only on one PDE. As we clarified above, our work does not pursue “Pretraining on multiple classes of PDEs”.
>
>
> > **Q2**. Pretrain different models for different PDEs
>
> 1. We provide extra experiments in **Figure 1 in our attached PDF response**. We can see that **joint pretraining can further improve the performance of fine-tuning on different PDEs**. We will include this figure in our camera-ready version.
> 2. Although joint pretraining can be further beneficial, experiments of **pretraining on a single PDE are still fundamental and necessary**. These should be studied before moving to joint pretraining and **cannot be ignored or underestimated**. These experiments help analyze the complexity or difficulty of learning each single PDE for neural operators.
> 3. **Recent works, like [4], also only adopted per-PDE pretraining, even though their work is claimed to be more related to SciML foundation models than ours**. Previous unsupervised pretraining works in computer vision also focused on a single dataset (like ImageNet).
>
>
> > **Q3**. Missing baselines
>
> On time-independent PDEs, we mainly focus on the FNO model. There are no publicly available checkpoints of FNO that are pretrained on natural images. Due to limited time during rebuttal, it is also challenging to pretrain an FNO ourselves on large-scale image datasets like ImageNet or JFT-300M. We will try to include this experiment in our camera-ready version.
>
> Meanwhile, on time-dependent PDEs, we studied Video-MAE because we followed previous work [5]. In [5], they also did not consider any image-pretrained MAE models for their claims.
>
>
> > **Q4**. "in-context learning" in Sec. 3.2
>
> We are happy to rename our method.
>
> Meanwhile, we have further corresponding comments:
> 1) In recent literature about large language models (LLMs), “context” is mostly referred to in downstream tasks, not in pretrained models, and we did not claim that our context is in our pretrained models;
> 2) In-context learning is not directly related to auto-regressive models, see [6];
> 3) Models with self-attention are also not learning anything during inference, since all model weights are fixed. Both self-attention and our models are extracting features during inference.
>
>
> > **Q5**. Accuracy and standard deviation of few-shot learning
>
> We report standard deviations in **Figure 3 in our attached PDF response**.
>
> The performance of neural operators in out-of-distribution (OOD) settings is known to be poor and challenging. See **Figure B.3 (f) of [4]**: performance on Helmholtz (“SYS-3”) is poor (both their errors and ours are close to 1), even with supervised fine-tuning on a few OOD samples.
>
>
> > **Q6**. Line 95: BERT does not leverage next token prediction for pretraining
>
> Thanks! We will update this sentence with a more precise description.
>
>
> > **Q7**. Line 142: Multiple benefits of pretraining are listed, but no references are provided
>
> As clearly mentioned in Line 148, these experiments about benefits of unsupervised pretraining are in Figure 4 in Sec. 4.1, which is in the main text and should not be ignored.
>
>
> > **Q8**. Line 158: I don't think any variant of BERT is scaled to one hundred billion parameters
>
> Thanks! We will cite more papers with models of billion-level parameters that share similar pretraining strategies as BERT.
>
>
> > **Q9**. Other issues.
>
> Thanks! We will fix these typos.
>
>
> > **Q10**. Can the pretrained model be applied to other PDEs which are not seen in pretraining?
>
> We provide experiments for fine-tuning on PDEs unseen during pretraining, shown in **Figure 2 in our attached PDF response**. Fine-tuning on unseen PDEs leads to worse performance, which is expected not only because they have different initial conditions but also due to their mismatched input dimensions. The input dimension of Poisson is four (forcing function and three diffusion coefficients), while the input dimension of Helmholtz is two (source function and the wavenumber). This mismatched input is documented in Section A.
>
>
> [1] “A Simple Framework for Contrastive Learning of Visual Representations” Chen et al. 2020
>
> [2] “Improved Baselines with Momentum Contrastive Learning” Chen et al. 2020
>
> [3] “Masked Autoencoders Are Scalable Vision Learners” He et al. 2021
>
> [4] “Towards Foundation Models for Scientific Machine Learning: Characterizing Scaling and Transfer Behavior“ Subramanian et al. 2023
>
> [5] “Multiple Physics Pretraining for Physical Surrogate Models” McCabe et al. 2023
>
> [6] “In-Context Operator Learning with Data Prompts for Differential Equation Problems” Liu et al. 2023

---

> > ### Author Response · Authors · 2024-08-09
> > **Fix a typo in our response**
> >
> > We would like to fix a typo in our response to your "**Q5**. Accuracy and standard deviation of few-shot learning".
> >
> > "Figure B.3 (f) of [4]" ==> "Figure **B.2** (f) of [4]".
> > We are referring to subplot (f) in "Figure B.2: Addressing (Q4)" of [4] (in Appendix B.2, at the very bottom of the paper).
> >
> > [4] “Towards Foundation Models for Scientific Machine Learning: Characterizing Scaling and Transfer Behavior“ Subramanian et al. 2023

---

> > > ### Comment · Reviewer_oUb1 · 2024-08-10
> > >
> > > I thank the authors for the detailed rebuttal. I still have several concerns:
> > >
> > > - While the authors argue that pretraining on a single PDE are still fundamental and necessary, this increase the computation cost to solve a single PDE. The author should report the total training time of pretraining and finetuning compared to the random initialization experiments for the readers to see the trade-off. I raised questions on the pretraining cost and finetuning cost in my original review but can not find the authors' response on this.
> > >
> > > - I thank the authors for trying to add MAE in the FNO experiments. However, at this stage, there is no relevant unsupervised learning baseline comparisons in the FNO experiments. As a reference, the existing work, _Multiple Physics Pretraining for Physical Surrogate Models_, show much more comprehensive baseline comparisons, e.g., in Table 1.
> > >
> > > - I disagree with the authors on _models with self-attention are also not learning anything during inference_ - if this is the case, why there should be the notion of "in-context learning"? Existing studies find that Transformers can _learn_ from the few-shot examples even if the weights are fixed [1-4]. This is in contrast to "extracting features" with the proposed model.
> > >
> > > [1] Akyürek, Ekin, et al. "What learning algorithm is in-context learning? investigations with linear models." arXiv preprint arXiv:2211.15661 (2022).
> > >
> > > [2] Von Oswald, Johannes, et al. "Transformers learn in-context by gradient descent." International Conference on Machine Learning. PMLR, 2023.
> > >
> > > [3] Zhang, Ruiqi, Spencer Frei, and Peter L. Bartlett. "Trained transformers learn linear models in-context." arXiv preprint arXiv:2306.09927 (2023).
> > >
> > > [4] Bai, Yu, et al. "Transformers as statisticians: Provable in-context learning with in-context algorithm selection." Advances in neural information processing systems 36 (2024).
> > >
> > > - The authors promise to _cite more papers with models of billion-level parameters that share similar pretraining strategies as BERT_. I'm not aware of any relevant model like this (i.e., based on masked language modeling). Can the authors kindly share some references?
> > >
> > > I maintain my ratings due to the concerns (the first two in particular).

---

> ### Author Response · Authors · 2024-08-10
> **Thank you very much for your reply!**
>
> We truly thank reviewer oUb1 for reading our response and providing the reply!
>
> Below are our responses to your further questions:
>
> > **Q1**. Pretraining cost and finetuning cost.
>
> 1. We reported our pretraining cost in our response to reviewer 6vSy (https://openreview.net/forum?id=MuPlJ9fT4b&noteId=9dXGDW9BRK). We apologize for this omitted response!
>   - Pretraining costs. FNO: 20 GPU hours. Video-MAE: 18 GPU hours. (GPU: A100)
>
> 2. For fine-tuning costs: FNO: 4 GPU hours. Video-MAE: 6 GPU hours. (GPU: A100).
>
> Meanwhile, we hope that reviewer oUb1 does not ignore our two other responses:
> 1) **Extra experimental results**: In Figure 1 in our PDF response. We can see that joint pretraining can further improve the performance of fine-tuning on different PDEs. We will include this figure in our camera-ready version.
> 2) **Previously published works also focused on single-dataset pretraining and did not compare the costs of pretraining/fine-tuning with those of data simulation/collection**: For example, [1] also only adopted per-PDE pretraining, even though their work is claimed to be more related to SciML foundation models than ours. Previous unsupervised pretraining works in computer vision also focused on a single dataset (like ImageNet).
>
>
> > **Q2**. “no relevant unsupervised learning baseline comparisons in the FNO experiments”
>
> Since we are the first to introduce unsupervised learning in SciML, there are indeed no previous related works (that also train FNO in an unsupervised manner) with which we can fairly compare.
>
>
> > **Q3**. “Multiple Physics Pretraining for Physical Surrogate Models, show much more comprehensive baseline comparisons in Table 1.”
>
> The reason McCabe et al. needed to compare with more baselines is that their work proposed both a new architecture and pretraining methods. They needed to demonstrate the benefits of both their architecture (by comparing with models with a comparable number of parameters) and their pretraining (by comparing with models trained on a single dataset).
>
> However, in our case, for each subplot in Figure 3: 1) If we compare with different models, we cannot draw meaningful conclusions because the architecture changes; 2) Since ours is the only work on unsupervised pretraining of neural operators, comparing with supervised pretrained models would be unfair.
>
> Instead, we chose the following approach: 1) To address different **architectures**, we studied both FNO and transformer models on different PDEs; 2) To address different **pretraining methods**, we included a vision-pretrained transformer. We believe these two aspects already cover multiple representative baselines that fulfill the purpose requested by reviewer oUb1.
>
>
> > **Q4**. In-context learning of transformers.
>
> We thank these references, all of them are well awared by we authors. We respect reviewer oUb1’s option. Our definition of “learning” is to update model parameters, and we are open to accepting other definitions.
>
>
> > **Q5**. Relevant model of billion-level parameters that share similar pretraining strategies as BERT.
>
> One example is the T5 model [2], which is of billion-level parameters and is pretrained with masked language modeling similar to BERT.
>
> Here we quote:
> * “we consider an analogous objective to BERT’s “masked language modeling” objective in” in their Sec. 3.
> * “we consider an objective inspired by the “masked language modeling” (MLM) objective used in BERT“ in their Sec. 3.3.1.
>
> We again thank reviewer oUb1's reply! We are happy to address any further concerns.
>
>
>
>
> [1] “Towards Foundation Models for Scientific Machine Learning: Characterizing Scaling and Transfer Behavior“ Subramanian et
> al. 2023
>
> [2] “Exploring the Limits of Transfer Learning with a Unified Text-to-Text Transformer” Raffel et al. 2019

---

> > ### Author Response · Authors · 2024-08-13
> > **Look forward to more discussions**
> >
> > Dear Reviewer oUb1,
> >
> > As the author-reviewer discussion period is nearing its end, and since other reviewers have actively engaged in discussions, we would greatly appreciate it if you could review our responses to your comments at your earliest convenience.
> >
> > This will allow us to address any further questions or concerns you may have before the discussion period concludes.
> >
> > If our responses satisfactorily address your concerns, we kindly ask you to consider revising your rating of our work.
> >
> > Thank you very much for your time and effort!
> >
> > Sincerely,
> >
> > The Authors of Submission #11275

---

> > > ### Comment · Reviewer_oUb1 · 2024-08-13
> > >
> > > I thank the authors for the response.
> > >
> > > - To support the claim on _even outperforming conventional vision-pretrained models_, I'm still expecting comparisons with a model pretrained on vision data in Fig 3a-3d. The comparisons do not need to be extremely large-scale (e.g., pretraining on JFT-300M).
> > >
> > > - I also share the concern of Reviewer oEEz on the effectiveness of the "in-context learning" method. And I still find the claim of "in-context learning" confusing. I agree with Reviewer oEEz that the authors need to focus the contributions on the pretraining strategy. I also suggest that the authors should consider renaming "in-context learning" in the paper as, for example, similarity-based inference with pretrained models.
> > >
> > > I will update my rating if the authors can promise to address these concerns in the revision.

---

> > > > ### Author Response · Authors · 2024-08-13
> > > > **Thank you very much for your reply!**
> > > >
> > > > We truly thank reviewer oUb1 for reading our response and providing the reply!
> > > >
> > > > > **Q1**. Model pretrained on vision data in Fig 3a-3d.
> > > >
> > > > Yes, after the rebuttal, we will have enough time to train our own 2D-FNO on ImageNet and a 3D-FNO on video datasets. Then we promise to add vision-trained FNO as another baseline to Poisson, Helmholtz, Reaction-Diffusion, and Navier-Stokes.
> > > >
> > > >
> > > > > **Q2**. Renaming "in-context learning"
> > > >
> > > > Thanks for this suggestion! We promise to change “ICL” to another name in our camera ready to avoid any confusion.

---

### Author Rebuttal · Authors · 2024-08-07

We deeply appreciate the feedback and suggestions from all four reviewers. We are pleased that **all four reviewers recognized** that our paper targets an **interesting and well-known challenge** in scientific machine learning (SciML).

We thank **all four reviewers** for acknowledging our main contribution, which is to **advance the pretraining method for SciML** and **reduce the number of labeled training samples**. This is an important research direction. We also thank reviewers oEEz, 6vSy, and u3i6 for confirming our comprehensive experiments on diverse PDEs, broad ranges of physical parameters, and real-world problems.

We address all questions and concerns in individual responses. Following the NeurIPS guidelines, we also attach a one-page PDF with figures of new experiments.

---

### Decision · Program_Chairs · 2024-09-25

**Decision:**

Accept (poster)

**Comment:**

The authors consider the problem of training neural operators to solve forward partial differential equation (PDE) problems in data-limited settings, motivated by the large data simulation cost incurred by these methods when generating a large training set of PDE solutions. They propose a self-supervised pretraining method derived from tasks popular in the computer vision community, and demonstrate the benefit of their method on a variety of PDE problems and real-world experimental data, compared to other self-supervised baselines. Since the majority of operator learning approaches require “supervised” data obtained by solving the forward PDE, most baselines they compare to are to not necessarily state-of-the-art (only SotA for the unsupervised setting, which has been less studied). The authors also show experiments on an "out-of-distribution" (OOD) task where PDE coefficient settings are changed at test time and an in-context learning strategy is used together with their pretrained models to improve performance relative to baselines, as in in-context operator learning.

After the rebuttal and discussion, reviewers’ scores for the paper increased, converging towards accept. The authors provided numerous clarifications on unclear details, especially methodological and experimental, in the paper during the rebuttal, as well as new experiments which convinced reviewers of the solidity of the paper’s methodology. Reviewers expressed some concerns about the competitiveness of the authors’ experimental baselines, but acknowledge that they are reasonable in the setting of self-supervised pretraining for operator learning that the authors consider, which has been relatively less explored than the supervised case. The AC concurs with the reviewers judgment on these points, seeing the paper as providing a nontrivial contribution to a potentially important research direction (especially in light of the clarifications of computational savings given in the response to u3i6’s review), and therefore recommends acceptance. The authors should not fail to incorporate all improvements in presentation and new experimental results recommended by the reviewers and presented during the rebuttal into the revision in a prominent way (i.e., not just in the appendices), especially regarding the proposed ICL method (per comments of oEEz and oUb1, the clear dependence on the in-context operator learning approach should be emphasized in Section 3.2), results on computational savings given to u3i6, and all clarifications of protocol made in the discussion, especially with u3i6.